environmental science/oceanography

microplastic, hadal, trench, microfibre, marine, pollution

**Author for correspondence:**
A. J. Jamieson
e-mail: alan.jamieson@ncl.ac.uk

# Microplastics and synthetic particles ingested by deep-sea amphipods in six of the deepest marine ecosystems on Earth

A. J. Jamieson[1], L. S. R. Brooks[1], W. D. K. Reid[1],
S. B. Piertney[2], B. E. Narayanaswamy[3] and T. D. Linley[1]

[1]Marine Sciences, School of Natural and Environmental Sciences, Newcastle University, Newcastle upon Tyne, Tyne and Wear NE1 7RU, UK
[2]Institute of Biological and Environmental Sciences, University of Aberdeen, Zoology Building, Tillydrone Avenue, Aberdeen AB24 2TZ, UK
[3]Scottish Association for Marine Science, Scottish Marine Institute, Oban, Argyll PA37 1QA, UK

AJJ, 0000-0001-9835-2909; WDKR, 0000-0003-0190-0425;
TDL, 0000-0002-6583-3105

While there is now an established recognition of microplastic pollution in the oceans, and the detrimental effects this may have on marine animals, the ocean depth at which such contamination is ingested by organisms has still not been established. Here, we detect the presence of ingested microplastics in the hindguts of Lysianassoidea amphipod populations, in six deep ocean trenches from around the Pacific Rim (Japan, Izu-Bonin, Mariana, Kermadec, New Hebrides and the Peru-Chile trenches), at depths ranging from 7000 m to 10 890 m. This illustrates that microplastic contaminants occur in the very deepest reaches of the oceans. Over 72% of individuals examined (65 of 90) contained at least one microparticle. The number of microparticles ingested per individual across all trenches ranged from 1 to 8. The mean and standard error of microparticles varied per trench, from $0.9 \pm 0.4$ (New Hebrides Trench) to $3.3 \pm 0.7$ (Mariana Trench). A subsample of microfibres and fragments analysed using FTIR were found to be a collection of plastic and synthetic materials (Nylon, polyethylene, polyamide, polyvinyl alcohol, polyvinylchloride, often with inorganic filler material), semi-synthetic (rayon and lyocell) and natural fibre (ramie). Notwithstanding, this study reports the deepest record of microplastic ingestion, indicating that anthropogenic debris is bioavailable to organisms at some of the deepest locations in the Earth's oceans.

# 1. Introduction

There is now an established appreciation of microplastic pollution in our oceans and the detrimental effects this has on marine organisms [1–3]. An estimated 322 million tons of plastic are produced annually [4], with more than 5 trillion plastic pieces weighing over 250 000 tons currently floating on the surface [5]. In 2010 alone, 4.8–12.7 million tons was released into the ocean and this is set to increase by an order of a magnitude by 2025 [6]. As such, plastics represent arguably the clearest indicator of mankind's detrimental impact on the oceans [7] and an obvious signature of the Anthropocene. A research priority is now to characterize the extent of microplastic and semi-synthetic fibre pollution in the oceans and the consequences this has on marine life. The investigation of microplastic ingestion by marine organisms has largely focused on shallow water habitats given the ease of sampling these locations yet we know very little about their ingestion in the deep sea [7–9]. This begs the questions of how pervasive and ubiquitous microplastic pollution is within the deep sea, and does it extend to full ocean depth?

The majority of plastic present in the oceans can be observed floating on the surface [9]. The degradation and fragmentation of plastics will ultimately result in sinking to the underlying deep-sea habitats, where opportunities for dispersal become ever more limited [7,9,10]. Marine plastic litter has now been observed in numerous locations in the deep sea [11–15]. The deepest recorded plastic item was plastic bag at 10 898 m in the Mariana Trench [15] while in the Ryukyu Trench off Japan at depths greater than 7000 m, discarded items were found with increasing frequency towards the trench axes [16]. This reflects the 'depocentre' function otherwise positively associated with surface-derived food supply [17].

Microplastics, defined as being between 0.1 µm and 5 mm [18], are of particular concern in marine environments because they may be similar or smaller in size to prey or particles selected for ingestion by marine organisms. Some microplastics are produced for industrial processes [19,20], while others have originated from the break-up of larger items through UV light and physical abrasion [20,21]. The size of microplastics makes them bioavailable, which facilitates entry into the food chain at various trophic levels and bioaccumulation [22–24].

Microplastic ingestion has been observed in a wide range of taxa including plankton [25], bivalves [26,27], crustaceans [28,29], echinoderms [8,9], fishes [30–34], elasmobranchs [35] and cetaceans [1,36]. The extent of the adverse effects on marine biota is not fully understood despite being known to negatively affect approximately 700 marine species, predominantly through ingestion, decreased nutrition from intestinal blockage or decreased mobility [3]. There is also the potential for plastics to act as a vector for pollutants including persistent organic pollutants (e.g. polychlorinated biphenyls) [37,38]. The downstream impacts at an ecosystem level on the physical and toxicological impacts of microplastic ingestion still remain unclear [38–40].

The major pathways for plastics to the oceans are diverse and range from river and estuary transport [41] to atmospheric fallout [42]. As a result, microplastics are observed globally in coastal [26,43], open ocean [44], pelagic [45], benthic [46] and deep-sea habitats [12,47,48]. There are only a few records of microplastics in deep-sea sediments [7,12,48] with the deepest point being 5768 m on the upper margins of the Kuril-Kamchatka Trench [12]. Currently, the deepest recorded occurrence of microplastic ingestion by deep-sea organisms is 2200 m depth in the North Atlantic [9] with no information about whether microplastics are being ingested by abyssal or hadal organisms. This means that we still do not know whether microplastics are ingested by the organisms that live at some of the deepest points in the ocean.

Given the range of transport pathways, the quantities produced and released each year, plus the estimates of the volume currently floating in the ocean, particularly in the large gyres, it is intuitive that the ultimate sink for this debris, in whatever size, is the deep sea [7]. Plastics reaching the massive expanse of the deep sea are ultimately contaminating an ecosystem we know far less about than the area from where it originates. This is especially the case in the hadal zone (6000–11 000 m depth [42]), which is the biozone composed largely of deep subduction zones, topographically isolated in large elongate trenches or depressions. The organisms living in these habitats are dependent on organic matter supplied from the surface [49], which, in turn, brings any adverse components, such as plastics and pollutants with it. For example, Jamieson *et al.* [50] have reported extraordinary bioaccumulation of persistent organic pollutants (POPs) in hadal fauna from deep subduction trenches in the Pacific Ocean. The deep sea is not only the ultimate sink for any material that descends from the surface but it is also inhabited by organisms well adapted to a low-food environment. Many deep-sea organisms, including amphipods, exhibit high trophic plasticity and have evolved diverse morphological and physiological adaptations to ensure feeding success at rare opportunities; therefore, in the presence of relatively new foreign bodies, the likelihood of ingestion is high [51].

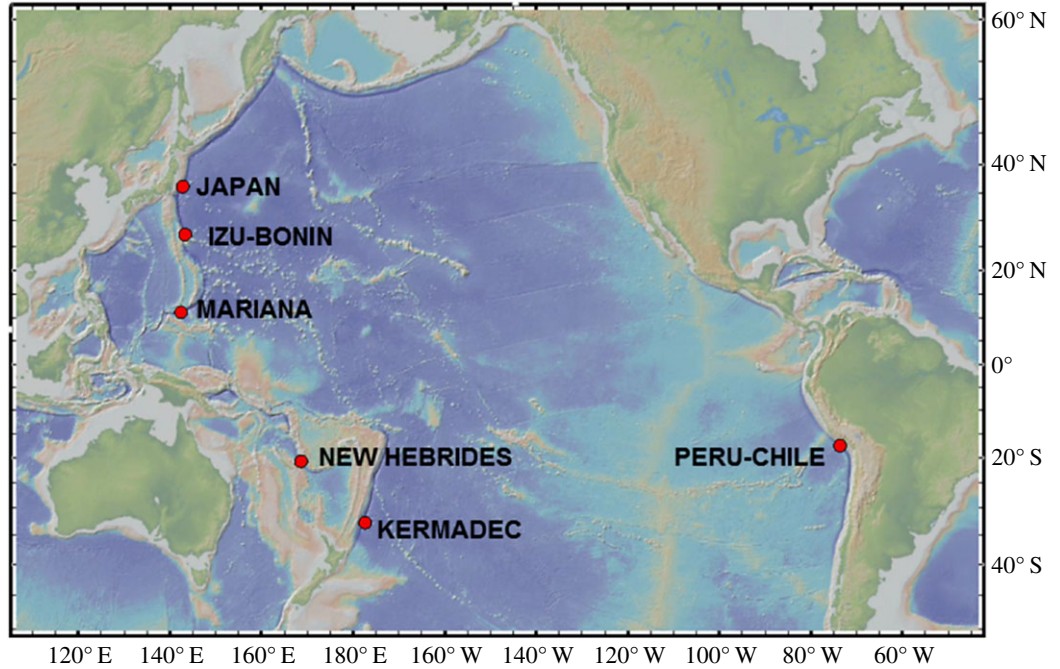

**Figure 1.** Locations of the six trenches around the Pacific rim where amphipods were sampled for microplastic ingestion. The sites include the Japan, Izu-Bonin and Mariana trenches in the northwest Pacific; the New Hebrides and Kermadec trenches in the southwest Pacific and the Peru-Chile Trench in the southeast Pacific.

The objective of this study was to examine the extent of microplastic and microfibre pollution across some of the deepest points of the ocean. Specifically, this study investigated the presence of ingested microplastic fibres and fragments in the hindgut of lysianassoid amphipods across multiple hadal trenches around the Pacific Rim. These included the Peru-Chile Trench in the Southeast Pacific, the New Hebrides and Kermadec trenches in the Southwest Pacific and the Japan, Izu-Bonin and Mariana trenches in the Northwest Pacific (figure 1). The latter contains the deepest point on the Earth, Challenger Deep at 10 890 m. The presence of microplastics at some or all these sites would demonstrate the reach of anthropogenic activity into evermore poorly understood and remote parts of the planet.

## 2. Methods

Three species of the lysianassoid amphipods (two *Hirondellea* sp. and *Eurythenes gryllus*; figure 2) were sampled across multiple cruises to the Japan, Izu-Bonin, Mariana, Kermadec, New Hebrides and Peru-Chile trenches between 2008 and 2017 (table 1). These trenches cover a wide spatial distribution within the Pacific Ocean and encompassed a depth range from approximately 7000 m to 10 890 m at the Challenger Deep, Mariana Trench and four depths were chosen within the Kermadec Trench (7014, 7884, 9053 and 9908 m). As such, a total of nine sites were examined.

The focal amphipod species were the dominant scavenging species in their respective trenches [53]. Ten individuals from each of the nine sites were examined. The samples were collected via small funnel traps (6 cm diameter by 30 cm length with an opening of approx. 2.5 cm) that were deployed on various Hadal-Lander vehicles [52], baited with locally sourced mackerel wrapped in a mesh to prevent bait consumption that could affect future studies. The mesh was either galvanized steel wire or bright yellow plastic. Furthermore, samples were taken only from the internal hindgut of the specimen to remove the possibility of contamination from substances consumed via the bait, wrap or from the lander. The ballast release mechanism on the Hadal-Lander featured a potential source of plastic microfibre in the form of a Dacron (synthetic polyester; polyethylene terephthalate) line that prior to 2010 was bright green and after 2010 was fluorescent yellow. These distinct colours meant that any similar coloured fibres found within the amphipod could be easily disregarded in the unlikely event they appeared in the hindgut. Upon retrieval from depth, the amphipods were stored in 70–99% ethanol in transparent plastic jars. Preservation of fauna in ethanol does not appear to significantly impact or degrade the microplastics [26].

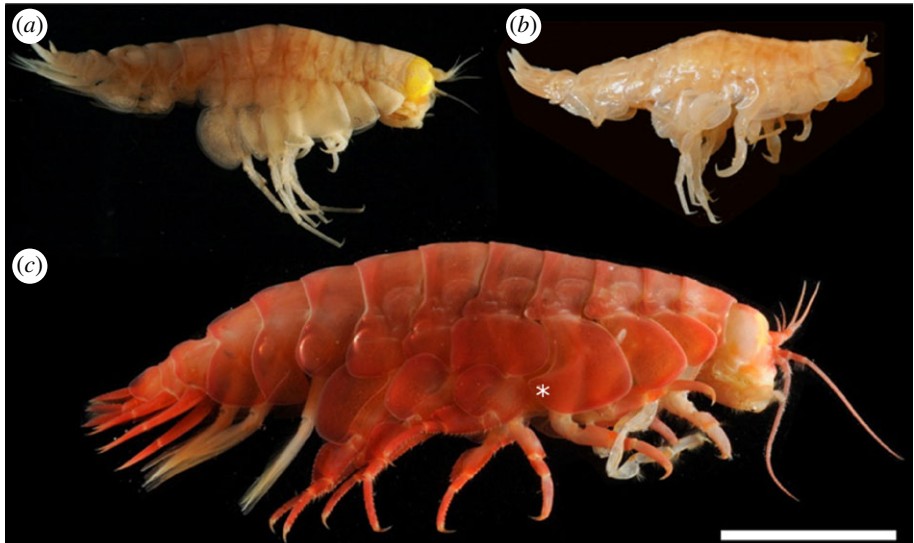

**Figure 2.** The three species of Lysianassoidea amphipods collected from six hadal trenches around the Pacific rim. (*a*) *Hirondellea gigas*, (*b*) *Hirondellea dubia* and (*c*) *Eurythenes gryllus*. Scale bar = 10 mm. * indicates position of coxa 4.

**Table 1.** Sampling locations of nine populations of Lysianassoidea amphipods across six Pacific hadal trenches: Japan (JT), Izu-Bonin (IBT), Mariana (MT), New Hebrides (NHT), Kermadec (KT) and Peru-Chile (PCT). The gears used to collect the amphipods were HL, Hadal-Lander, versions A, B and C; OBS1, Obulus lander version 1; Latis, Latis lander [52].

| trench | region | depth (m) | date | cruise | latitude | longitude | gear | species |
|---|---|---|---|---|---|---|---|---|
| JT | NW | 7703 | 30.09.08 | KH0803 | 36.24933 | 142.81683 | HL-A | *H. gigas* |
| IBT | NW | 9316 | 18.03.09 | KT0903 | 27.34983 | 143.31483 | HL-A | *H. gigas* |
| MT | NW | 10 890 | 29.01.17 | SY1615 | 11.36683 | 142.42986 | HL-C | *H. gigas* |
| NHT | SW | 6948 | 21.11.13 | KAH1310 | −20.6485 | −168.6138 | HL-C | *H. dubia* |
| KT | SW | 7014 | 28.11.11 | KAH1109 | −32.75958 | −177.24091 | OBS1 | *H. dubia* |
| KT | SW | 7884 | 29.11.11 | KAH1109 | −32.61641 | −177.35822 | Latis | *H. dubia* |
| KT | SW | 9053 | 21.02.12 | KAH1202 | −31.9785 | −177.3885 | Latis | *H. dubia* |
| KT | SW | 9908 | 30.11.11 | KAH1109 | −32.02657 | −177.37083 | Latis | *H. dubia* |
| PCT | SE | 7050 | 10.09.10 | SO0209 | −17.4245 | −73.61683 | HL-B | *E. gryllus* |

Precautionary measures were put in place to prevent any airborne and liquid contamination within the laboratory. Surfaces, glassware and dissection equipment were rinsed with acetone, followed by a final rinse with type one ultra-pure water directly before use. To prevent solvent contamination, all liquids were filtered using Whatman No. 540 filter paper [54]. Laboratory coats and nitrile gloves were worn throughout. Dissection and identification occurred within a laminar flow hood cabinet (Thorflow EDF600) to restrict airborne contamination. Samples were sealed prior to removal from the laminar flow hood for digestion. Procedural control blanks, done concurrently with samples, showed no contamination although the fibrous filter membrane showed partly loose, clear fibres on some samples, hence clear fibres were excluded from results. We did not find any white fibres that may have been contamination from the white laboratory coat worn during sample preparation.

## 2.1. Fibre and fragment identification

Under laminar flow, amphipods were individually dissected to remove the hindgut; defined as the body cavity posterior to Coxa 4. The hindgut weight was recorded before samples were digested, following [52], with 10% potassium hydroxide (KOH) incubated over a 48 h period at 40°C within a grade C fume vent. The volume of KOH used was at least three times greater than that of individual gut

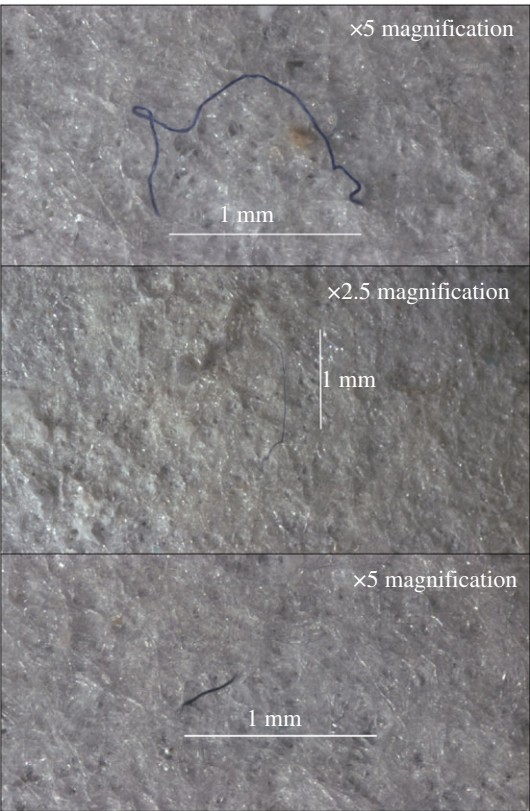

**Figure 3.** A selection of microfibre examples found within amphipod hindgut samples from 10 890 m in the Mariana Trench.

weight [34]. KOH has been shown to be a suitable solution to dissolve the guts of marine fauna, leaving the majority of microplastics unaffected [55].

After digestion samples were left to cool before being filtered through Whatman No. 541 filter paper, filters were then transferred onto a Petri dish for stereomicroscopic analysis (Nikon ocular 40x, Intralux 4000-1). The abundance of observed microparticles (those particles which had not been digested) was recorded and categorized by colour and shape (e.g. figure 3) [56,57]. The samples were then wrapped in muffled tin foil and transferred to a photolab where representative digital images were taken (Canon EOS 1300D DSLR) to provide visual information on colour and differences in shape across the nine sites.

A subsample of fibres ($n = 15$) spanning all trenches were analysed by Fourier-transform infrared spectrophotometer (FTIR; IR Tracer-100, Shimadzu, Japan) connected to an automatic infrared microscope (AIM-9000, Shimadzu, Japan) at the Shimadzu UK Ltd Laboratory Facility in Milton Keynes. Individual fibres were manually removed and transferred to the surface of FTIR reflective slides (Kevley Technologies, Ohio) (which provide a suitable background for reflectance) or transferred to a Specac DC3 diamond cell and compressed for transmission measurements (with background scans being taken through the diamond adjacent to the sample). The fibres presented in the results were analysed by transmission as this provided the most reliable results. The fibre was observed using the wide-field camera to identify possible locations for further investigation and the measurements were made in transmittance or reflectance mode (50 scans over approx. 20 s) using the wide-band MCT (mercury cadmium telluride) detector. For each fibre, three points were scanned and the results were compared to those in the Shimadzu materials library for matches or closest similarity. Some of the fibres which showed unusual structure were scanned in several places to reveal more about their chemical composition.

# 3. Results

Microparticles of man-made synthetic or semi-synthetic fibres and fragments were found in the hindgut of amphipods at all nine sites (figure 4a). The frequency of ingestion varied between 50 and 100% of

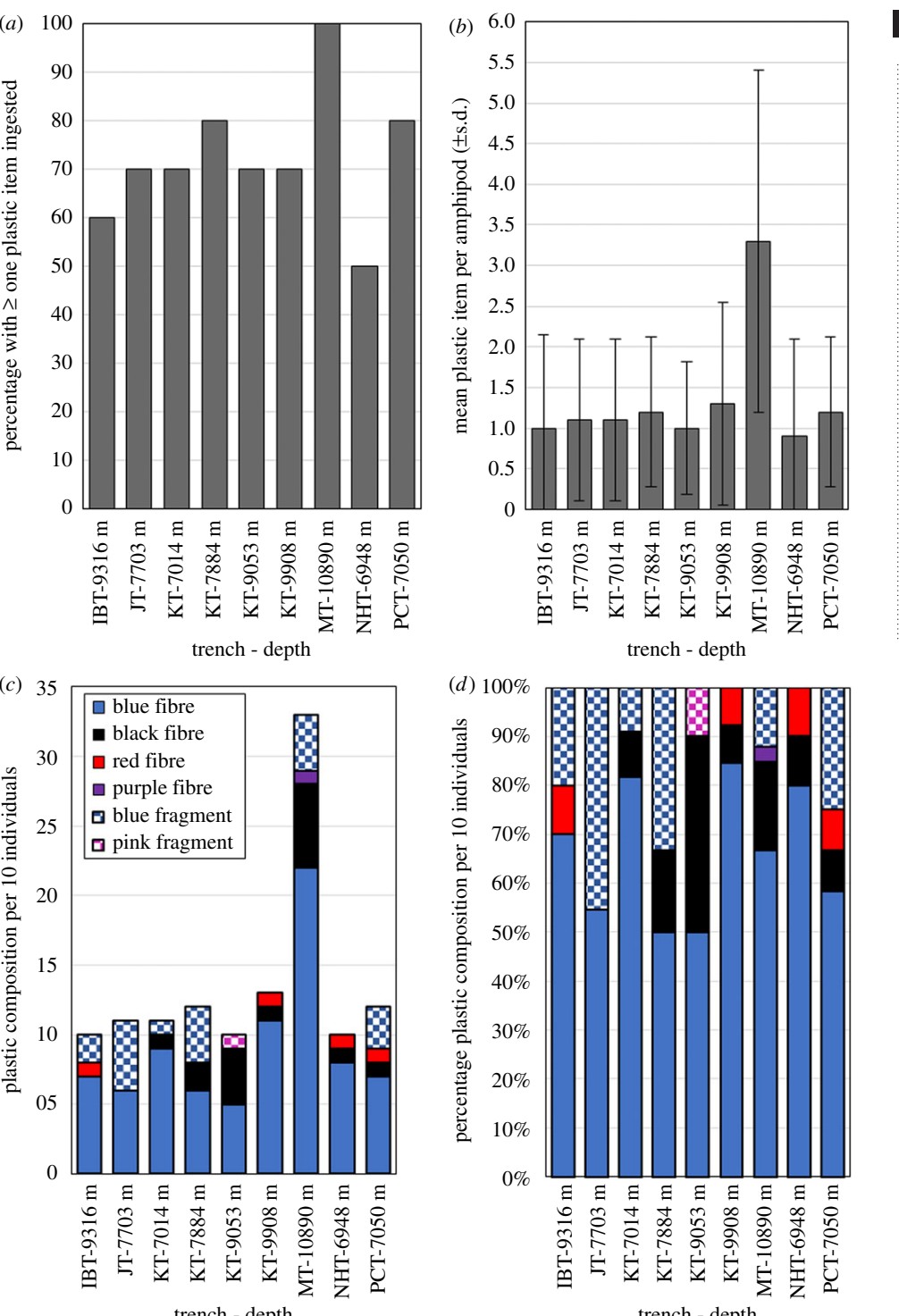

**Figure 4.** (*a*) Percentage of amphipods with at least one ingested particle item; (*b*) the mean (±s.d.) number of items per individual; (*c*) composition of colour and type and (*d*) composition colour and type of particle expressed as percentage. All plotted against site (and depth), *n* = 10. Abbreviations for the sites are: JT, Japan Trench; IBT, Izu-Bonin Trench; MT, Mariana Trench; NHT, New Hebrides Trench; KT, Kermadec Trench; and PCT, Peru-Chile Trench.

amphipods from a given site; the lowest being the New Hebrides Trench (50%) and the highest the Mariana Trench (100%). Of the 90 individual amphipods examined, 65 individuals (approx. 72%) contained at least one microfibre or fragment. The mean and standard error (s.e.) of the number of items ingested per individual of all amphipods sampled in all trenches was 1.34 ± 1.1 (range: one to eight items per individual). The New Hebrides Trench amphipods contained the lowest mean number

**Table 2.** Results of the FTIR analysis on fibre material across six Pacific hadal trenches: Japan (JT), Izu-Bonin (IBT), Mariana (MT), New Hebrides (NHT), Kermadec (KT) and Peru-Chile (PCT).

| trench | depth (m) | material | description |
|---|---|---|---|
| JT | 7703 | lyocell | blue fibre |
| IBT | 9316 | polyester reinforced cotton; rayon | twisted blue fibre |
| IBT | 9316 | polyethylene | degraded fibre, red |
| MT | 10 890 | low density polyethylene film with inorganic filler | dark/black fibre |
| MT | 10 890 | ramie | blue fibre |
| MT | 10 890 | ramie | blue fibre |
| NHT | 6948 | unidentified polyvinyl | dark/blue fibre |
| NHT | 6948 | PA with inorganic filler | dark/black fibre |
| KT | 7014 | lyocell | black fibre |
| KT | 9908 | unidentified plastic | black fibre |
| KT | 7884 | unidentified plastic, but very close to PVAL or PVC with inorganic filler | dark/blue fragment |
| KT | 9908 | ramie | blue fibre |
| KT | 9053 | nylon 12 | black/dark fibre |
| PCT | 7050 | polyester core with polyethylene coating | black fibre |
| PCT | 7050 | polyethylene with inorganic filler | black fibre |

of microparticles ($0.9 \pm 0.4$) and the Marina Trench had the highest ($3.3 \pm 0.7$) (figure 4*b*). There was no relationship between the number of microparticles and depth in the Kermadec Trench amphipods (Kruskal–Wallis $\chi^2 = 0.23$, d.f. $= 3$, $p = 0.97$).

A total of 122 ingested microparticles were identified and were categorized into fibres and fragments (figure 4*c*). Fibres were found within every trench and appeared in 84% of amphipods, whereas the occurrence of fragments was lower and appeared in only 16% of amphipods. No fragments were found in the New Hebrides Trench amphipods.

Using a crude colour-based categorization, the most prevalent items ingested were blue fibres (66%) with all amphipods sampled from the Marina Trench containing at least one of these. The next most prevalent items ingested were blue fragments (16%) followed by black fibres (13%), red fibres (4%), pink fragments (less than 1%) and purple fibres (less than 1%). However, the FTIR analysis revealed that these fibre and fragment groupings did not correspond to a single material type but rather a variety of materials (table 2). Six of the 15 items analysed using FTIR were semi-synthetic cellulosic fibres, rayon and lyocell, the natural fibre ramie that are used in products such as textiles. The rest included synthetic polymers such as Nylon, polyethylene (PE), polyamide (PA) or unidentified polyvinyls closely resembling polyvinyl alcohol (PVAL) or polyvinylchloride (PVC) and with most including an inorganic filler material. One fibre found in the Peru-Chile Trench at 7050 m was clearly a polyethylene-coated strand of polyester. None of the 15 subsamples were found to be naturally occurring.

## 4. Discussion

The salient finding of this study is that man-made microfibres and fragments, including microplastics, were found in the hindguts of amphipods from six of the deepest parts of the Earth's oceans, including within the deepest area of the Mariana Trench, at Challenger Deep. Plastic has been present at hadal depths for the last couple of decades [15] but, to the best of our knowledge, this is the first record of microplastics being ingested by hadal organisms. Therefore, microplastics are bioavailable in the hadal zone and ingested by one of the most important and dominant scavenging fauna in the deep sea at a similar frequency (72%) to crustaceans in coastal water habitats [28,29].

Previous studies have found that microplastics ingested by deep-sea invertebrates down to 2200 m in the North Atlantic [9], 611 m in the equatorial mid-Atlantic [8] and 1062 m in southwest Indian

Ocean [8]. The species ingesting microplastics include the echinoderms *Ophiomusium lymani*, *Hymenaster pellucidus* (North Atlantic) [9] and an unknown species of holothurian (southwest Indian Ocean) [8]; a crustacean (unknown hermit crab) from the southwest Indian Ocean [8]; and a mollusc (*Colus jeffreysianus*) from the North Atlantic [9]. As with the amphipods in this study, these species are all deposit feeders or are predatory [8,9]. It is not clear whether these trophic guilds are more susceptible to microplastic ingestion in the deep sea than filter feeders [8] or whether there are toxicological implications as microplastics breakdown [37]. This can only be tested with a wider range of species from different trophic guilds with accompanying microplastic concentrations from sediments and the water column.

The six trenches are bathymetrically and geographically isolated by large distances. The distance between the Japan Trench, in the Northern Hemisphere, and the Kermadec Trench, in the Southern Hemisphere, is approximately 8640 km, and between the Peru-Chile Trench in the Southeast Pacific and the trenches in the northwest Pacific is over 15 000 km. The distances highlight the geographical extent in the distribution of microplastics and synthetic particles that are ingested at full ocean depths. It is difficult to ascertain why the amphipods have different numbers of microparticles in their hindguts among these six trenches. The mechanisms transporting microplastics and synthetic fibres to the deep sea are likely to be similar at all the locations. These include sinking of large plastics (greater than 5 mm) from the surface waters and subsequent fragmentation at depth [7,12,15,48]; direct sinking of microplastic that are not adhered to other particles; sinking of microplastics in association with marine snow [18,48]; and the downward transport of large and microplastics in the stomachs of vertically migrating pelagic organisms and marine carrion [31,45]. The temporal mismatch among sampling the trenches is a confounding factor when explaining why there are differences in observed numbers of microparticles in the amphipod stomachs. The differences may be related to the duration of time that plastics have accumulated in the area rather than whether areas accumulate more plastic in the surface or deep water and if there are regional differences in the mechanisms that transport plastics to the deep sea. However, given our sampling occurred from 2008 onwards, it indicates that microplastics were ingested by amphipods for at least the past decade, providing an important baseline to monitor subsequent change.

The crude colour-based categorization is consistent with findings in surface waters where fibres dominate and account for a high proportion of microplastics [58]. The source and mechanism by which these microplastics are released into the marine environment is varied and includes airborne transport, terrestrial sources, e.g. synthetic fibres from washing clothes which enters the marine environment through sewage [59–62], direct release of fibres through maritime activities, e.g. fishing [21] and fragmentation of larger plastic debris. Blue fibres were the most prevalent microparticles ingested in the Pacific hadal amphipods which is consistent with other studies [44,58]. Furthermore, in Pacific subsurface water, black, red and purple fibres [58] are also prevalent; all of these colours were found ingested in Pacific hadal amphipods in this study. However, it is clear from the FTIR analysis and previous works that the colour-based categorization is not an adequate method to identify whether a microparticle is indeed of plastic origin [63]. The range of plastic found in the hindguts of the amphipods included PE, PA and polyvinyls resembling PVAL or PVC but we also found other synthetic polymers that are not plastics (e.g. ramie, lyocell). PE, PA and polyester have all been identified in the guts of other deep-sea organisms albeit at much shallower depths [8,9].

The presence of microplastics in the hindgut of amphipods indicates the possibility of trophic transfer to higher trophic levels within the hadal environment. Trophic transfer of microplastics is known from other marine organisms including from *Mytilus edulis* to *Carcinus maenas* [22] and between mesozooplankton to higher level macrozooplankton [23]. These studies were conducted under experimental conditions using high concentrations of microplastics, but their results indicate the possibility of microplastics transferring among individuals [22,23]. Amphipods are known prey for larger hadal taxa such as decapods [64], other predatory amphipods [65] and fish such as liparids and ophidiids [66–68]. Once the microplastics enter the hadal food chain, there is a strong possibility that they will be locked into a perpetual cycle of trophic transfer. This is because amphipods scavenge on marine carrion which includes those fish and decapods from surface waters as well as those living at similar depths that potentially are also their predators [68,69]. At depths greater than 8000 m, amphipods consume a combination of surface-derived marine detritus and carrion, and other species of amphipod [51], which again suggests the likelihood of inevitable trophic cycling of microplastics at these depths. The extent to which deep-sea amphipods can disperse microplastics across the seafloor is currently unclear. This is because their digestion and defecation rates are currently unknown.

# 5. Conclusion

The results of this study demonstrate that man-made fibres including microplastics are ingested by lysianassoid amphipods at the deepest location of all the Earth's oceans. Microplastic ingestion occurred in all trenches, indicating they are bioavailable within hadal environments. We hypothesize that the physical impacts known in shallower ecosystems as a result of microplastic ingestion [4] are likely to occur within hadal populations. Plastics are being ingested, culminating in bioavailability in an ecosystem inhabited by species we poorly understand, cannot observe experimentally and have failed to obtain baseline data for prior to contamination. This study reports the deepest record of microplastic ingestion, indicating it is highly likely there are no marine ecosystems left that are not impacted by plastic pollution.

Data accessibility. Our data are supplied here as electronic supplemental information.

Authors' contribution. A.J.J. participated in all sampling at sea, designed and built all the sampling equipment, and with S.B.P. was principal investigator or co-principal investigator on the grants that supported the specimen collection. A.J.J. devised this project and with W.K.D.R. supervised L.S.R.B.'s laboratory work. A.J.J., L.S.R.B., W.K.D.R. and S.B.P. interpreted the results and all authors contributed to the writing of the manuscript. All authors gave final approval for publication.

Competing interests. The authors declare no competing interests.

Funding. Funding for the laboratory work and analysis was from Newcastle University internal support. This work was supported by the 2007–2010 HADEEP project, funded by the Nippon Foundation (2009765188) and the Natural Environmental Research Council (NE/E007171/1). The 2011–2013 Kermadec Trench sampling was supported by the TOTAL Foundation (France) through the projects 'Multi-disciplinary investigations of the deepest scavengers on Earth' (2010–2012) and 'Trench Connection' (2013–2015). The Mariana samples were derived from the 'FISH2017' expedition (RV *Shinyo-Maru* SY1615) supported by the Tokyo University for Marine Science and Technology.

Acknowledgements. We thank the captain, crew and company of the research expeditions who assisted in the collection of the amphipods between 2008 and 2017, namely the Japanese *Hakuho-Maru*, *Tansei Maru* and *Shinyo-Maru*, the German *Sonne* and the RV *Kaharoa* in New Zealand. The assistance of David Whitaker and Peter McParlin from The School of Marine Science and Technology at Newcastle University are much appreciated. We are extremely grateful to Bob Keighley and Dan Parnaby at Shimadzu UK Limited for facilitating the FTIR analysis and access to their material database. We also thank Heather Stewart from the British Geological Survey for calculating the distances between trenches.

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
