## [Reviewer comments · Royal Society Open Science]

Review History

RSOS-170589.R0 (Original submission)

Review form: Reviewer 1

Is the manuscript scientifically sound in its present form?

No

Are the interpretations and conclusions justified by the results?

No

Is the language acceptable?

Yes

Is it clear how to access all supporting data?

No

Do you have any ethical concerns with this paper?

No

Have you any concerns about statistical analyses in this paper?

Yes

Recommendation?

Major revision is needed (please make suggestions in comments)

Comments to the Author(s)

Review of the article entitled "Microplastic ingestion in six of the deepest marine ecosystems on Earth"

The present study investigated the presence of microplastics in amphipod sampled in six deep-sea ecosystem. The authors extracted the microplastic from organisms with a digestion protocol then they took pictures of the microplastics and concluded that microplastics are present and ingested by amphipods at the deepest location of all the world's ocean.

Overall comments

This article is potentially interesting and timely. The data collected from an unexplored area of Pacific Ocean are of high scientific relevance. However, there are some major issues that prevent the acceptance of the manuscript in the present form and need to be addressed:

1. the study refers to "microplastic", but lacks of analytical (chemical or spectroscopic) analysis, which is absolutely necessary to confirm that the filaments they found are made of plastic and not of other materials (as often the case when made of cellulose or rayon or viscose). The fibres and fragments were indeed identified in terms of colour and shape from the material extracted from the animal.
2. the manuscript needs to be carefully revised / supported for some definitions/assumptions as some of them could have important scientific implications :
 - microplastic: the authors only specify the upper limit (≤ 5 mm)
 - refs for the ingestion of microplastic by other marine organisms such as deep-sea fish and invertebrate (e.g., polychaetes)
 - you take 3000 m as the maximum depth reached by the bathyal zone but this could be 4000 m depending on the definitions
 - refs for the different mechanisms of microplastic export to deep sea (e.g marine snow and biofouling)
3. The results are not clearly illustrated. The histogram in Figure 4c is not clear and intuitive. The percentage relative to the amphipods is not represented and apparently comes from the blue. I would recommend to include an additional figure illustrating the different percentages of the different fibres and fragments to be consistent with the discussion.

Concluding, I think that this article needs a throughout revision, including an in depth analysis of the plastic to confirm the results presented, and should provide a clear presentation of the results obtained before being considered for publication.

Specific comments**Introduction**

Line 39. Please, add a reference

Line 66, Please add a references
Line 78. Please add a reference
Line 81. Please add a reference
Line 83. Please add a reference

Methods

Line 101. Add a reference here

Results

Line 146. Here you used "identified", but no specific analysis was made thus you cannot say that is microplastic.

Line 152. Please, add a graph

Discussion

Line 183. Please add a reference

Results

Line 214, Please add a reference that state that blue fibres are derived from fishing nets

Line 215. Please add a reference for shallower ecosystems

Table and Figure

Table 1. Geographic coordinates should be provided in degrees

Figure 2. Please provide size range

Figure 3. Again please provide the scale, and include in the legend of the magnification used.

Review form: Reviewer 2

Is the manuscript scientifically sound in its present form?

Yes

Are the interpretations and conclusions justified by the results?

No

Is the language acceptable?

Yes

Is it clear how to access all supporting data?

Not Applicable

Do you have any ethical concerns with this paper?

No

Have you any concerns about statistical analyses in this paper?

No

Recommendation?

Major revision is needed (please make suggestions in comments)

Comments to the Author(s)

Dear Authors,

Thank you for presenting an interesting study. Investigating the interaction of deep-benthic organisms with microplastics has a great significance as the deep-sea has been highlighted as the eventual sink for microplastics.

I have found areas within the manuscript that should be addressed and i feel these changes will greatly improve the study. I have also taken time to critically look at the references included and have suggested some changes.

Specific comments: (using authors line numbers)

Abstract

- Line 11. I think it is time to move away from GROWING. Microplastics are now recognized worldwide as a form of pollution. Please amend accordingly
- Line 12: detrimental, only in the lab, in the main body text some more information on this could be added to support its inclusion in the abstract
- Line 23: change negligence to debris

Introduction

- Line 26: GROWING same as introduction
- Line 27: please address the references and update accordingly. [1-3] would be better replaced with a review to represent the breadth of the subject area and current knowledge. For example, GESAMP released a through review in 2016, which is an update and extension of Lusher 2015.
- Line 28: [4] is not a suitable reference, it is a fish paper. find a better reference
- Line 29 and 30: both references used here [5,6] need to be clearly stated that they are estimations based on limited empirical evidence and as such should be used with caution.
- Line 31: remove this reference and include reference to the Anthropocene,

Zalasiewicz, J., Waters, C.N., Wolfe, A.P., Barnosky, A.D., Cearreta, A., Edgeworth, M., Ellis, E.C., Fairchild, I.J., Gradstein, F.M., Grinevald, J. and Haff, P., 2017. Making the case for a formal Anthropocene Epoch: an analysis of ongoing critiques. *Newsletters on Stratigraphy*, 50(2), pp.205-226.

- Line 41: use a more recent reference. Woodall et al. might be suitable
- Line 41: the opportunities for dispersal are not the only reason plastics reach the deep sea, i would like to see other reasons for plastic accumulating in the deep sea, settling rates, density, biofouling etc. Some of the recent publications highlight and discuss these methods.
- Line 52: change reference [17] this is a terrestrial and aquatic study and not relevant to the deep sea
- Line 53: curious to why the word RAIN is used here, if the discussion of MPs route to the deep sea is expanded it may be relevant but at the moment it seems out of place
- Line 58: reference [56] is a methods paper i would recommend including a study such as Paul-Pont et al. 2016. is reference [27] accepted? if not please include a peer-reviewed publication.
- Line 62: reference [31] should be updated to be a MP study rather than general plastic debris, suggest GESAMP 2016
- Line 65; reference [37] should be replaced with a pelagic study, not on fish, suggest Lusher et al. 2014 or Enders et al. 2016 both are from the Atlantic, and Enders is middle of the ocean basin. Reference [38] should be replaced with a more up to date reference, there are many recent sediment papers. [40] is a macropalstic paper and suggest it is omitted from this list

Methods

-The methods used are sound as written but present a significant flaw in this research. There is no confirmation of the visually identified plastics. As it stands this study presents POTENTIAL

microplastics. As recent research has highlighted, visual identification is not recommended on its own, and it needs to be supported by further techniques. I urged the authors to address this significant error. I am highly opposed to publishing anything that does not even attempt to confirm the authors "visual " identification skills. A subset 10% of potential particles would improve interpretation of visual analysis greatly.

- I would like to some annotations to the images of the amphipods to show the location of the hindgut

Results

Line 146- remove individual

Line 148- remove sampled

Line 149- remove sampled

Discussion

Line 170: reference [38] is a very old study, please update with recent advances in the literature

Line 168-181- i would like to see some discussion on potential for airbourne contamination, this section would also be greatly supported by inclusion of chemical analysis of potential microplastics. For example, If Nylon is identified, then the authors could infer possible sources.

Line 186: i am not convinced that POPs and associated chemicals to plastics are relevant to this manuscript as the authors do not study the effects, its inclusion would be relevant if better introduced. I feel that more discussion on sources and routes of MPs to the deep sea sediment is more important.

Line 188-195: ref [15 and 62] is outdated, research has actually shown that plastics might be less of an exposure pathway than first thought I suggest the authors read and refer to more recent literature. there are some interesting discussions highlighted in GESAMP and more recent reviews. At the moment, I feel the authors are drawing a tedious link between chemicals, plastics and the deep sea.

Line 198- It is important to note that these studies are from laboratory exposure not the marine environment

Line 219 replace negligence with debris

Figures

Figure2. if possible i would love to see an indication of where the hind gut is

Figure3. please include a typical fragment

Figure4. would be interesting if a) was included on the map of the locations. could consider pie charts for presence absence percentage

c) minor error in the key., Black Fibre needs F to be f

Decision letter (RSOS-170589.R0)

02-Aug-2017

Dear Dr Jamieson:

Manuscript ID RSOS-170589 entitled "Microplastic ingestion in six of the deepest marine ecosystems on Earth" which you submitted to Royal Society Open Science, has been reviewed. The comments from reviewers are included at the bottom of this letter.

In view of the criticisms of the reviewers, the manuscript has been rejected in its current form. However, a new manuscript may be submitted which takes into consideration these comments. A critical matter that both reviewers raise is the need to characterise fully the plastics found in your study. Without such characterisation your interesting study will not have the impact it might otherwise have.

Please note that resubmitting your manuscript does not guarantee eventual acceptance, and that your resubmission will be subject to peer review before a decision is made.

Your resubmitted manuscript should be submitted by 30-Jan-2018. If you are unable to submit by this date please contact the Editorial Office.

Sincerely,
Alice Power
Editorial Coordinator
Royal Society Open Science

on behalf of
Jon Blundy, Royal Society Open Science
openscience@royalsociety.org

Reviewers' Comments to Author:

Reviewer: 1

Comments to the Author(s)

Review of the article entitled "Microplastic ingestion in six of the deepest marine ecosystems on Earth"

The present study investigated the presence of microplastics in amphipod sampled in six deep-sea ecosystem. The authors extracted the microplastic from organisms with a digestion protocol then they took pictures of the microplastics and concluded that microplastics are present and ingested by amphipods at the deepest location of all the world's ocean.

Overall comments

This article is potentially interesting and timely. The data collected from an unexplored area of Pacific Ocean are of high scientific relevance. However, there are some major issues that prevent the acceptance of the manuscript in the present form and need to be addressed:

1. the study refers to "microplastic", but lacks of analytical (chemical or spectroscopic) analysis, which is absolutely necessary to confirm that the filaments they found are made of plastic and not

of other materials (as often the case when made of cellulose or rayon or viscose). The fibres and fragments were indeed identified in terms of colour and shape from the material extracted from the animal.

2. the manuscript needs to be carefully revised / supported for some definitions/assumptions as some of them could have important scientific implications :

- microplastic: the authors only specify the upper limit (≤ 5 mm)
- refs for the ingestion of microplastic by other marine organisms such as deep-sea fish and invertebrate (e.g., polychaetes)
- you take 3000 m as the maximum depth reached by the bathyal zone but this could be 4000 m depending on the definitions
- refs for the different mechanisms of microplastic export to deep sea (e.g marine snow and biofouling)

3. The results are not clearly illustrated. The histogram in Figure 4c is not clear and intuitive. The percentage relative to the amphipods is not represented and apparently comes from the blue. I would recommend to include an additional figure illustrating the different percentages of the different fibres and fragments to be consistent with the discussion.

Concluding, I think that this article needs a throughout revision, including an in depth analysis of the plastic to confirm the results presented, and should provide a clear presentation of the results obtained before being considered for publication.

Specific comments

Introduction

Line 39. Please, add a reference

Line 66, Please add a references

Line 78. Please add a reference

Line 81. Please add a reference

Line 83. Please add a reference

Methods

Line 101. Add a reference here

Results

Line 146. Here you used "identified", but no specific analysis was made thus you cannot say that is microplastic.

Line 152. Please, add a graph

Discussion

Line 183. Please add a reference

Results

Line 214, Please add a reference that state that blue fibres are derived from fishing nets

Line 215. Please add a reference for shallower ecosystems

Table and Figure

Table 1. Geographic coordinates should be provided in degrees

Figure 2. Please provide size range

Figure 3. Again please provide the scale, and include in the legend of the magnification used.

Reviewer: 2

Comments to the Author(s)

Dear Authors,

Thank you for presenting an interesting study. Investigating the interaction of deep-benthic organisms with microplastics has a great significance as the deep-sea has been highlighted as the eventual sink for microplastics.

I have found areas within the manuscript that should be addressed and i feel these changes will greatly improve the study. I have also taken time to critically look at the references included and have suggested some changes.

Specific comments: (using authors line numbers)

Abstract

- Line 11. I think it is time to move away from GROWING. Micropalstics are now recognized worldwide as a form of pollution. Please amend accordingly
- Line 12: detrimental, only in the lab, in the main body text some more information on this could be added to support its inclusion in the abstract
- Line 23: change negligence to debris

Introduction

- Line 26: GROWING same as introduction
- Line 27: please address the references and update accordingly. [1-3] would be better replaced with a review to represent the breadth of the subject area and current knowledge. For example, GESAMP released a through review in 2016, which is an update and extension of Lusher 2015.
- Line 28: [4] is not a suitable reference, it is a fish paper. find a better reference
- Line 29 and 30: both references used here [5,6] need to be clearly stated that they are estimations based on limited empirical evidence and as such should be used with caution.
- Line 31: remove this reference and include reference to the Anthropocene,

Zalasiewicz, J., Waters, C.N., Wolfe, A.P., Barnosky, A.D., Cearreta, A., Edgeworth, M., Ellis, E.C., Fairchild, I.J., Gradstein, F.M., Grinevald, J. and Haff, P., 2017. Making the case for a formal Anthropocene Epoch: an analysis of ongoing critiques. *Newsletters on Stratigraphy*, 50(2), pp.205-226.

- Line 41: use a more recent reference. Woodall et al. might be suitable
- Line 41: the opportunities for dispersal are not the only reason plastics reach the deep sea, i would like to see other reasons for plastic accumulating in the deep sea, settling rates, density, biofouling etc. Some of the recent publications highlight and discuss these methods.
- Line 52: change reference [17] this is a terrestrial and aquatic study and not relevant to the deep sea
- Line 53: curious to why the word RAIN is used here, if the discussion of MPs route to the deep sea is expanded it may be relevant but at the moment it seems out of place
- Line 58: reference [56] is a methods paper i would recommend including a study such as Paul-Pont et al. 2016. is reference [27] accepted? if not please include a peer-reviewed publication.
- Line 62: reference [31] should be updated to be a MP study rather than general plastic debris, suggest GESAMP 2016
- Line 65; reference [37] should be replaced with a pelagic study, not on fish, suggest Lusher et al. 2014 or Enders et al. 2016 both are from the Atlantic, and Enders is middle of the ocean basin.

Reference [38] should be replaced with a more up to date reference, there are many recent sediment papers. [40] is a macropalstic paper and suggest it is omitted from this list

Methods

-The methods used are sound as written but present a significant flaw in this research. There is no confirmation of the visually identified plastics. As it stands this study presents POTENTIAL microplastics. As recent research has highlighted, visual identification is not recommended on its own, and it needs to be supported by further techniques. I urged the authors to address this significant error. I am highly opposed to publishing anything that does not even attempt to confirm the authors "visual " identification skills. A subset 10% of potential particles would improve interpretation of visual analysis greatly.

- I would like to some annotations to the images of the amphipods to show the location of the hindgut

Results

Line 146- remove individual

Line 148- remove sampled

Line 149- remove sampled

Discussion

Line 170: reference [38] is a very old study, please update with recent advances in the literature

Line 168-181- i would like to see some discussion on potential for airbourne contamination, this section would also be greatly supported by inclusion of chemical analysis of potential microplastics. For example, If Nylon is identified, then the authors could infer possible sources.

Line 186: i am not convinced that POPs and associated chemicals to plastics are relevant to this manuscript as the authors do not study the effects, its inclusion would be relevant if better introduced. I feel that more discussion on sources and routes of MPs to the deep sea sediment is more important.

Line 188-195: ref [15 and 62] is outdated, research has actually shown that plastics might be less of an exposure pathway than first thought I suggest the authors read and refer to more recent literature. there are some interesting discussions highlighted in GESAMP and more recent reviews. At the moment, I feel the authors are drawing a tedious link between chemicals, plastics and the deep sea.

Line 198- It is important to note that these studies are from laboratory exposure not the marine environment

Line 219 replace negligence with debris

Figures

Figure2. if possible i would love to see an indication of where the hind gut is

Figure3. please include a typical fragment

Figure4. would be interesting if a) was included on the map of the locations. could consider pie charts for presence absence percentage

c) minor error in the key., Black Fibre needs F to be f

Author's Response to Decision Letter for (RSOS-170589.R0)

See Appendix A.

RSOS-172333.R0

Review form: Reviewer 2

Is the manuscript scientifically sound in its present form?

No

Are the interpretations and conclusions justified by the results?

No

Is the language acceptable?

Yes

Is it clear how to access all supporting data?

Not Applicable

Do you have any ethical concerns with this paper?

No

Have you any concerns about statistical analyses in this paper?

Yes

Recommendation?

Major revision is needed (please make suggestions in comments)

Comments to the Author(s)

Review: RSOS-172333

Microfibre ingestion in size of the deepest marine ecosystems on earth

Dear Authors,

Thank you for producing this manuscript. You have investigated the presence of microfibres/particles ingestion by amphipods in the deep sea. This information is timely and provides a new insight into deep sea contamination. The introduction is well references, although some references can be updated. The object of the study is clearly stated. I am concerned about the number of individuals per site, although I do understand the difficulties in collecting specimens from such depth. I would urge the authors to therefore reduce the speculation of site differences in the results due to the low number of individuals. There are two few specimens per sites to carry out sufficient statistical analysis. I lack size descriptions of particles identified. Please include more discussion on other papers looking at deep sea organisms. The discussion has a large text on toxicological effects, I feel too much weight is given here, considering the study does not focus on toxicity. I would like to see more information on the effect of KOH, and how the sites are not comparable. Please expand.

Specific comments (line numbers supplied by authors)

Title: update to be microplastics or microparticles to truly reflect the results

Introduction

Line 30: please update this value, 322 was the latest value from Plastic Europe in 2015.

Line 30: litters is a very colloquial term, I would suggest the authors state occurs/is present.

Line 57: by definition fibres are not primary microplastics, they are produced through the breakdown. I would also advise the authors to avoid terminology such as primary and secondary

microplastics. There are three types of microplastics. Those that are produced for use in the microscale, those that breakdown during use, and those that breakdown after use/once they are environmental contaminants.

Line 57: reference 18 is not needed here.

Methods

Line 107: 10 individuals per site is very low for replicates. I would therefore highlight that any differences between sites cannot be classed as significant. Further replicates are needed.

Line 108: remove baited. (is repeated in 109, so not needed twice)

Line 110: Change to ... could affect "further study."

Line 113: Question: do the authors know how fast digestion is in this species, could digestion happen during capture? I would like to see some discussion around this point.

Line 118: Question: Are the authors aware of any previous research that shows the % of ethanol that affects plastic?

Line 141: why was only 15 particles subjected to FTIR? Does this follow MSFD and NOAA that recommend 10% of overall subsample based on size?

Line 149: Why were both transmittance and reflectance used? They can provide different results, ATR is better at distinguishing between semi- cellulosic particle.

Results

Line 155: fibres AND fragments

Line 165: 122 items in total. Were all particles plastics/semi-synthetic, how many were rejected based on the FTIR results?

Line 176: ramie, correct spelling please

Line 181: Please discuss the potential effect of KOH on particles.

Discussion

Line 188: please include other papers on deep water organisms. Taylor etc.

Line 193-195: Due to the date differences, are the samples comparable? I lack discussion here.

Line 214-218: this study does not show these. I suggest the authors make it clear

Conclusion

Lin 233: here you have used world's whereas earlier in the document you have used World's. Please be consistent

Line 235: which factors? Please include,. Would like to see these factors discussed in more details in the discussion.

Line 236-237 should be removed as no a key result from this study

Line 238: This study does not show plastics are culminating and accumulating. Should not be a conclusion.

Review form: Reviewer 3

Is the manuscript scientifically sound in its present form?

No

Are the interpretations and conclusions justified by the results?

No

Is the language acceptable?

Yes

Is it clear how to access all supporting data?

Yes

Do you have any ethical concerns with this paper?

No

Have you any concerns about statistical analyses in this paper?

No

Recommendation?

Reject

Comments to the Author(s)

The manuscript presents observations of microplastics from amphipods sampled from hadal depths. It covers six locations across the general Pacific region. From the observations authors conclude 'it is likely that all marine ecosystems ... have been impacted by anthropogenic debris'. I suggest that the authors did not require their new data to make this conclusion, therefore need to reconsider what their data bring to the crowded field of microplastics research.

Specific comments follow

Abstract

Line 13-14 untrue- Ubiquity has been shown across geography and depths. Agreed not the depths presented in this paper, how everywhere microplastics have been looked for they have been found. Suggest rephrasing this statement.

Line 22 Plastic polymer names should be given in full instead of the abbreviations

Line 25-26, this is not new and the data in this paper are not required for this hypothesis- see main comment

Keywords

Spelling 'microfiber'

Introduction

Line 36-41- Suggest removing, not relevant to the paper

Line 53- See Koelman et al 2016 and Bakir et al 2016 for discussion on this. It is not as certain as authors suggest. It is important to qualify this statement such as 'Plastic could act as vectors...'. Koelmans, A. A., Bakir, A., Burton, G. A., & Janssen, C. R. (2016). Microplastic as a vector for chemicals in the aquatic environment: critical review and model-supported reinterpretation of empirical studies. *Environmental science & technology*, 50(7), 3315-3326.

Bakir, A., O'Connor, I. A., Rowland, S. J., Hendriks, A. J., & Thompson, R. C. (2016). Relative importance of microplastics as a pathway for the transfer of hydrophobic organic chemicals to marine life. *Environmental pollution*, 219, 56-65.

Line 55 ref 16 is a review and does not support this statement in fact it states 'Hydrophobic contaminants are enriched on microplastics, but the available experimental results and modelling approaches indicate that the transfer of sorbed pollutants by microplastics is not likely to contribute significantly to bioaccumulation of these pollutants.'

Line 63-Might be useful to add in this sentence that the negative impacts documented to date are those from physical mechanisms of the microplastics, as these are the impacts listed.

Line 72- Fischer et al already show microplastics in hadal sediments. Why is it not appropriate to extrapolate plastic presence to greater depth and organism ingestion to those inhabiting those depths? The mechanisms for microplastic sinking given in the papers cited, continue through the depths of the ocean.

Methods

Line 118- Was the container cleaned?

Line 123- What was lab coat made from? Where there any other controls about clothing worn in the lab and during sample collection?

Results

Line 175-181 Suggest deleting. Not sure in the value of suggesting which products the microplastics could have come from. Polymers are used over a vast range of products and it is

not possible to track microplastics to a source. Maybe a simple statement that says a wide varied of polymers were observed and therefore they likely come from a wide variety of products.

Discussion

An important point for the discussion is that light microscope screening into categories does not work, as stated lines 172-173.

Line 196- Suggest deleting as the crude sorting process was not confirmed with FTIR. Therefore discussion results based on a classification they disprove is not useful. In addition I am unsure how useful it is to compare between studies just using plastic colour as they could have been of any material seen or unseen in the new data set.

Line 209 - See comments associated with line 53. You really need to state that most lab studies use concentrations and models that are not representative in the wild see review by Galloway et al (2017) for discussion.

Galloway, T. S., Cole, M., & Lewis, C. (2017). Interactions of microplastic debris throughout the marine ecosystem. *Nature ecology & evolution*, 1(5), 0116.

Line 222 Trophic transfer conclusions are based on lab studies using very high concentrations. Therefore the discussion needs to reflect this.

Conclusion

Line 235- Due to limited sampling not sure it is an appropriate to conclude higher ingestion rate in the Mariana Trench.

Line 236 Suggest amending this sentence. Physical impacts have been shown not chemical ones. Ref 39 does not support this statement.

Line 242 This is not a new conclusion see ref 7 cited in this manuscript

References

Line 295, ref 8 spelling 'the'

Line 394 ref 48 Italics for scientific name

Line 403 ref 52 check capitalisation

Figure 3 legend- Suggest change '3' to 'three'

Table 1- Give details on gear OBS and Latis

Decision letter (RSOS-172333.R0)

23-Feb-2018

Dear Dr Jamieson:

Manuscript ID RSOS-172333 entitled "Microfibre ingestion in six of the deepest marine ecosystems on Earth" which you submitted to Royal Society Open Science, has been reviewed. The comments from reviewer(s) are included at the bottom of this letter.

In view of the criticisms of the reviewer(s), I must decline the manuscript for publication in Royal Society Open Science at this time. However, a new manuscript may be submitted which takes into consideration these comments. I note that this is already a resubmission and that many of the original issues regarding your manuscript have not been satisfactorily addressed. Specifically, these include a tendency to over-hype the new data, in terms of their novelty and significance and, related, a failure to fully cite other, recent pertinent literature. That said, I find your manuscript to be an important addition to a growing body of evidence for the pernicious influence of plastics into the deepest parts of the ocean ecosystem. However, for your manuscript to be published in RSOS, it is essential to recognise the limitations of your work (sample size, habitat, etc) and not over-egg its significance. This is what reviewers have consistently and reasonably sought, but I feel you have failed to do. In inviting you to resubmit I am offering another chance to make the required changes and then face further review. If you are unwilling

(or unable) to make such changes then it may be that your work is better suited to another journal.

Please note that resubmitting your manuscript does not guarantee eventual acceptance, and that your resubmission will be subject to re-review by the reviewer(s) before a decision is rendered.

You will be unable to make your revisions on the originally submitted version of your manuscript. Instead, revise your manuscript using a word processing program and save it on your computer.

You may also click the below link to start the resubmission process (or continue the process if you have already started your resubmission) for your manuscript. If you use the below link you will not be required to login to ScholarOne Manuscripts.

*** PLEASE NOTE: This is a two-step process. After clicking on the link, you will be directed to a webpage to confirm. ***

https://mc.manuscriptcentral.com/rsos?URL_MASK=b27d689726424cdebbea29446355623d

Because we are trying to facilitate timely publication of manuscripts submitted to Royal Society Open Science, your resubmitted manuscript should be submitted by 23-Aug-2018. If you are unable to submit by this date please contact the Editorial Office for options.

Please note that Royal Society Open Science will introduce article processing charges for all new submissions received from 1 January 2018. Charges will also apply to papers transferred to Royal Society Open Science from other Royal Society Publishing journals, as well as papers submitted as part of our collaboration with the Royal Society of Chemistry (<http://rsos.royalsocietypublishing.org/chemistry>). If your manuscript is submitted and accepted for publication after 1 Jan 2018, you will be asked to pay the article processing charge, unless you request a waiver and this is approved by Royal Society Publishing. You can find out more about the charges at <http://rsos.royalsocietypublishing.org/page/charges>. Should you have any queries, please contact openscience@royalsociety.org.

I look forward to a resubmission.

on behalf of Jon Blundy (Subject Editor)
openscience@royalsociety.org

Reviewer comments to Author:

Reviewer: 2

Comments to the Author(s)

Review: RSOS-172333

Microfibre ingestion in size of the deepest marine ecosystems on earth

Dear Authors,

Thank you for producing this manuscript. You have investigated the presence of microfibres/particles ingestion by amphipods in the deep sea. This information is timely and provides a new insight into deep sea contamination. The introduction is well references, although some references can be updated. The object of the study is clearly stated. I am concerned about the number of individuals per site, although I do understand the difficulties in collecting specimens from such depth. I would urge the authors to therefore reduce the speculation of site differences in the results due to the low number of individuals. There are two few specimens per sites to carry out sufficient statistical analysis. I lack size descriptions of particles identified. Please include more discussion on other papers looking at deep sea organisms. The discussion has a large text on toxicological effects, I feel too much weight is given here, considering the study does not focus on toxicity. I would like to see more information on the effect of KOH, and how the sites are not comparable. Please expand.

Specific comments (line numbers supplied by authors)

Title: update to be microplastics or microparticles to truly reflect the results

Introduction

Line 30: please update this value, 322 was the latest value from Plastic Europe in 2015.

Line 30: litters is a very colloquial term, I would suggest the authors state occurs/is present.

Line 57: by definition fibres are not primary microplastics, they are produced through the breakdown. I would also advise the authors to avoid terminology such as primary and secondary microplastics. There are three types of microplastics. Those that are produced for use in the microscale, those that breakdown during use, and those that breakdown after use/once they are environmental contaminants.

Line 57: reference 18 is not needed here.

Methods

Line 107: 10 individuals per site is very low for replicates. I would therefore highlight that any differences between sites cannot be classed as significant. Further replicates are needed.

Line 108: remove baited. (is repeated in 109, so not needed twice)

Line 110: Change to ... could affect "further study."

Line 113: Question: do the authors know how fast digestion is in this species, could digestion happen during capture? I would like to see some discussion around this point.

Line 118: Question: Are the authors aware of any previous research that shows the % of ethanol that affects plastic?

Line 141: why was only 15 particles subjected to FTIR? Does this follow MSFD and NOAA that recommend 10% of overall subsample based on size?

Line 149: Why were both transmittance and reflectance used? They can provide different results, ATR is better at distinguishing between semi- cellulosic particle.

Results

Line 155: fibres AND fragments

Line 165: 122 items in total. Were all particles plastics/semi-synthetic, how many were rejected based on the FTIR results?

Line 176: ramie, correct spelling please

Line 181: Please discuss the potential effect of KOH on particles.

Discussion

Line 188: please include other papers on deep water organisms. Taylor etc.

Line 193-195: Due to the date differences, are the samples comparable? I lack discussion here.

Line 214-218: this study does not show these. I suggest the authors make it clear

Conclusion

Lin 233: here you have used world's whereas earlier in the document you have used World's.

Please be consistent

Line 235: which factors? Please include,. Would like to see these factors discussed in more details in the discussion.

Line 236-237 should be removed as no a key result from this study

Line 238: This study does not show plastics are culminating and accumulating. Should not be a conclusion.

Reviewer: 3

Comments to the Author(s)

The manuscript presents observations of microplastics from amphipods sampled from hadal depths. It covers six locations across the general Pacific region. From the observations authors conclude 'it is likely that all marine ecosystems ... have been impacted by anthropogenic debris'. I suggest that the authors did not require their new data to make this conclusion, therefore need to reconsider what their data bring to the crowded field of microplastics research.

Specific comments follow

Abstract

Line 13-14 untrue- Ubiquity has been shown across geography and depths. Agreed not the depths presented in this paper, how everywhere microplastics have been looked for they have been found. Suggest rephrasing this statement.

Line 22 Plastic polymer names should be given in full instead of the abbreviations

Line 25-26, this is not new and the data in this paper are not required for this hypothesis- see main comment

Keywords

Spelling 'microfiber'

Introduction

Line 36-41- Suggest removing, not relevant to the paper

Line 53- See Koelman et al 2016 and Bakir et al 2016 for discussion on this. It is not as certain as authors suggest. It is important to qualify this statement such as 'Plastic could act as vectors...'. Koelmans, A. A., Bakir, A., Burton, G. A., & Janssen, C. R. (2016). Microplastic as a vector for chemicals in the aquatic environment: critical review and model-supported reinterpretation of empirical studies. *Environmental science & technology*, 50(7), 3315-3326.

Bakir, A., O'Connor, I. A., Rowland, S. J., Hendriks, A. J., & Thompson, R. C. (2016). Relative importance of microplastics as a pathway for the transfer of hydrophobic organic chemicals to marine life. *Environmental pollution*, 219, 56-65.

Line 55 ref 16 is a review and does not support this statement in fact it states 'Hydrophobic contaminants are enriched on microplastics, but the available experimental results and modelling approaches indicate that the transfer of sorbed pollutants by microplastics is not likely to contribute significantly to bioaccumulation of these pollutants.'

Line 63-Might be useful to add in this sentence that the negative impacts documented to date are those from physical mechanisms of the microplastics, as these are the impacts listed.

Line 72- Fischer et al already show microplastics in hadal sediments. Why is it not appropriate to extrapolate plastic presence to greater depth and organism ingestion to those inhabiting those depths? The mechanisms for microplastic sinking given in the papers cited, continue through the depths of the ocean.

Methods

Line 118- Was the container cleaned?

Line 123- What was lab coat made from? Where there any other controls about clothing worn in the lab and during sample collection?

Results

Line 175-181 Suggest deleting. Not sure in the value of suggesting which products the microplastics could have come from. Polymers are used over a vast range of products and it is not possible to track microplastics to a source. Maybe a simple statement that says a wide varied of polymers were observed and therefore they likely come from a wide variety of products.

Discussion

An important point for the discussion is that light microscope screening into categories does not work, as stated lines 172-173.

Line 196- Suggest deleting as the crude sorting process was not confirmed with FTIR. Therefore discussion results based on a classification they disprove is not useful. In addition I am unsure how useful it is to compare between studies just using plastic colour as they could have been of any material seen or unseen in the new data set.

Line 209 - See comments associated with line 53. You really need to state that most lab studies use concentrations and models that are not representative in the wild see review by Galloway et al (2017) for discussion.

Galloway, T. S., Cole, M., & Lewis, C. (2017). Interactions of microplastic debris throughout the marine ecosystem. *Nature ecology & evolution*, 1(5), 0116.

Line 222 Trophic transfer conclusions are based on lab studies using very high concentrations. Therefore the discussion needs to reflect this.

Conclusion

Line 235- Due to limited sampling not sure it is an appropriate to conclude higher ingestion rate in the Mariana Trench.

Line 236 Suggest amending this sentence. Physical impacts have been shown not chemical ones. Ref 39 does not support this statement.

Line 242 This is not a new conclusion see ref 7 cited in this manuscript

References

Line 295, ref 8 spelling 'the'

Line 394 ref 48 Italics for scientific name

Line 403 ref 52 check capitalisation

Figure 3 legend- Suggest change '3' to 'three'

Table 1- Give details on gear OBS and Latis

Author's Response to Decision Letter for (RSOS-172333.R0)

See Appendix B.

RSOS-180667.R0

Review form: Reviewer 1

Is the manuscript scientifically sound in its present form?

Yes

Are the interpretations and conclusions justified by the results?

Yes

Is the language acceptable?

Yes

Is it clear how to access all supporting data?

Not Applicable

Do you have any ethical concerns with this paper?

No

Have you any concerns about statistical analyses in this paper?

No

Recommendation?

Accept as is

Comments to the Author(s)

The authors generally did a good job in amending the ms and provided a convincing response to all queries, as detailed here below:

1. Number of individual per site and statistical analysis: The authors justified the reason for the reduced number of samples. The study focuses more on the presence or absence of microplastic in organisms. FT-IR analysis have been added clarified in the text.
2. Comparability of the sites: the authors have answered in a comprehensive manner, limiting themselves to highlighting the individual results in the different sites, specifying that they do not know what the phenomena responsible for these different accumulations could be.
3. Include more discussion on other studies looking at deep-sea organisms: the authors did not include Choy et al. 2013
4. Discussion on toxicological effects: this part of the discussion has been omitted in the amended removed
5. Effects of potassium hydroxide (KOH): the authors have have included more detailed information on the treatment with KOH and added a reference that explains more about the treatment

Specific comments: all the comments have been taken into consideration by the authors, except for the line 113 where authors express the impossibility to test how fast the digestion is in the different species.

I'm happy with the present version and I recommend its publication as it is

Review form: Reviewer 2

Is the manuscript scientifically sound in its present form?

Yes

Are the interpretations and conclusions justified by the results?

No

Is the language acceptable?

Yes

Is it clear how to access all supporting data?

Yes

Do you have any ethical concerns with this paper?

No

Have you any concerns about statistical analyses in this paper?

Yes

Recommendation?

Accept with minor revision (please list in comments)

Comments to the Author(s)

Thank you for addressing my previous concerns. I have a few more minor comments

In your response to: Introduction Line 30: please update this value, 322 was the latest value from Plastic Europe in 2015.

You stated that: We were unable to find this value on Plastic Europe website. In light of this, we will stick with the value in Erickson et al. 2014 (10.1371/journal.pone.0111913) rather than placing in an unreferenced source.

Using data from three years ago is inappropriate. Here is the reference. See page 16. All scientific research has presented this data from earlier years, I see no reason why the authors cannot/refuse to use this data

<https://www.plasticseurope.org/en/resources/publications/274-plastics-facts-2017>

I reiterate that 10 individuals per site is not an appropriate number, there is reference in the literature that 20-50 individuals are required to give a good description of contamination loads. Especially in areas as remote as the trenches. I highly urge the authors to increase the numbers of individuals at each site, especially when they say there are more individuals to use for analysis. This will have a better impact on the research community and will not be looked at as "another" presence and absence paper.

Line 113: Question: do the authors know how fast digestion is in this species, could digestion happen during capture? I would like to see some discussion around this point.

Response: We do not know how fast digestion is in these species. Feeding studies are extremely difficult in hadal species because the amphipods arrive dead on deck as a result of decompression mortality.

I thank the authors for this response. Such discussion would be useful in the document. Digestion times are highlighted in coastal studies, and even staytong the ansers are not clear but furture research would be needed would be suffice for the discussion.

Decision letter (RSOS-180667.R0)

18-Oct-2018

Dear Dr Jamieson,

The Subject Editor assigned to your paper ("Microplastics and synthetic particles ingested by deep-sea amphipods in six of the deepest marine ecosystems on Earth") has now received

comments from reviewers. We would like you to revise your paper in accordance with the referee and Associate Editor suggestions which can be found below (not including confidential reports to the Editor). Please note this decision does not guarantee eventual acceptance.

Please submit a copy of your revised paper before 10-Nov-2018. Please note that the revision deadline will expire at 00.00am on this date. If we do not hear from you within this time then it will be assumed that the paper has been withdrawn. In exceptional circumstances, extensions may be possible if agreed with the Editorial Office in advance. We do not allow multiple rounds of revision so we urge you to make every effort to fully address all of the comments at this stage. If deemed necessary by the Editors, your manuscript will be sent back to one or more of the original reviewers for assessment. If the original reviewers are not available we may invite new reviewers.

When submitting your revised manuscript, you must respond to the comments made by the referees and upload a file "Response to Referees" in "Section 6 - File Upload". Please use this to document how you have responded to each of the comments, and the adjustments you have made. In order to expedite the processing of the revised manuscript, please be as specific as possible in your response.

- Ethics statement

- Data accessibility

<http://datadryad.org/submit?journalID=RSOS&manu=RSOS-180667>

- Competing interests

- Authors' contributions

- Acknowledgements

- Funding statement

Please note that Royal Society Open Science charge article processing charges for all new submissions that are accepted for publication. Charges will also apply to papers transferred to Royal Society Open Science from other Royal Society Publishing journals, as well as papers submitted as part of our collaboration with the Royal Society of Chemistry (<http://rsos.royalsocietypublishing.org/chemistry>). If your manuscript is newly submitted and subsequently accepted for publication, you will be asked to pay the article processing charge, unless you request a waiver and this is approved by Royal Society Publishing. You can find out more about the charges at <http://rsos.royalsocietypublishing.org/page/charges>. Should you have any queries, please contact openscience@royalsociety.org.

on behalf of Prof. Jon Blundy (Subject Editor)
openscience@royalsociety.org

Associate Editor Comments to Author:

The referees have made broadly favourable assessments of your resubmitted manuscript; however, a number of concerns remain, and the Editors consider that a revision is appropriate to allow and encourage you to submit responses and revisions to these concerns. Thank you for your submission.

Reviewer comments to Author:

Reviewer: 2

Comments to the Author(s)

Thank you for addressing my previous concerns. I have a few more minor comments

In your response to: Introduction Line 30: please update this value, 322 was the latest value from Plastic Europe in 2015.

You stated that: We were unable to find this value on Plastic Europe website. In light of this, we will stick with the value in Erickson et al. 2014 (10.1371/journal.pone.0111913) rather than placing in an unreferenced source.

Using data from three years ago is inappropriate. Here is the reference. See page 16. All scientific research has presented this data from earlier years, I see no reason why the authors cannot/refuse to use this data

<https://www.plasticseurope.org/en/resources/publications/274-plastics-facts-2017>

I reiterate that 10 individuals per site is not an appropriate number, there is reference in the literature that 20-50 individuals are required to give a good description of contamination loads. Especially in areas as remote as the trenches. I highly urge the authors to increase the numbers of individuals at each site, especially when they say there are more individuals to use for analysis. This will have a better impact on the research community and will not be looked at as "another" presence and absence paper.

Line 113: Question: do the authors know how fast digestion is in this species, could digestion happen during capture? I would like to see some discussion around this point.

Response: We do not know how fast digestion is in these species. Feeding studies are extremely difficult in hadal species because the amphipods arrive dead on deck as a result of decompression mortality.

I thank the authors for this response. Such discussion would be useful in the document. Digestion times are highlighted in coastal studies, and even staytong the ansers are not clear but furture research would be needed would be suffice for the discussion.

Reviewer: 1

Comments to the Author(s)

The authors generally did a good job in amending the ms and provided a convincing response to all queries, as detailed here below:

1. Number of individual per site and statistical analysis: The authors justified the reason for the reduced number of samples. The study focuses more on the presence or absence of microplastic in organisms. FT-IR analysis have been added clarified in the text.
2. Comparability of the sites: the authors have answered in a comprehensive manner, limiting themselves to highlighting the individual results in the different sites, specifying that they do not know what the phenomena responsible for these different accumulations could be.
3. Include more discussion on other studies looking at deep-sea organisms: the authors did not include Choy et al. 2013
4. Discussion on toxicological effects: this part of the discussion has been omitted in the amended removed
5. Effects of potassium hydroxide (KOH): the authors have have included more detailed information on the treatment with KOH and added a reference that explains more about the treatment

Specific comments: all the comments have been taken into consideration by the authors, except for the line 113 where authors express the impossibility to test how fast the digestion is in the different species.

I'm happy with the present version and I recommend its publication as it is

Author's Response to Decision Letter for (RSOS-180667.R0)

See Appendix C.

RSOS-180667.R1 (Revision)

Review form: Reviewer 1

Is the manuscript scientifically sound in its present form?

Yes

Are the interpretations and conclusions justified by the results?

Yes

Is the language acceptable?

Yes

Is it clear how to access all supporting data?

Yes

Do you have any ethical concerns with this paper?

No

Have you any concerns about statistical analyses in this paper?

No

Recommendation?

Accept with minor revision (please list in comments)

Comments to the Author(s)

Review of the article "Microplastic and synthetic particles ingested by deep-sea amphipods in six of the deepest marine ecosystems on Earth"

Reviewer: 1

The authors did a good job in amending the ms and provided a convincing response to all queries, except for the line 113 where authors declared the impossibility to test how fast the digestion is in the different species. I think there are some data available and the authors should make a further effort in this direction. The other minor issues have been answered in a comprehensive manner. Overall, however, I'm happy with the present version and I recommend its publication after this minor change.

Decision letter (RSOS-180667.R1)

28-Nov-2018

Dear Dr Jamieson:

On behalf of the Editors, I am pleased to inform you that your Manuscript RSOS-180667.R1 entitled "Microplastics and synthetic particles ingested by deep-sea amphipods in six of the deepest marine ecosystems on Earth" has been accepted for publication in Royal Society Open Science subject to minor revision in accordance with the referee suggestions. Please find the referees' comments at the end of this email.

The reviewers and Subject Editor have recommended publication, but also suggest some minor revisions to your manuscript. Therefore, I invite you to respond to the comments and revise your manuscript.

- Ethics statement

- Data accessibility

If you wish to submit your supporting data or code to Dryad (<http://datadryad.org/>), or modify your current submission to dryad, please use the following link:
<http://datadryad.org/submit?journalID=RSOS&manu=RSOS-180667.R1>

- Competing interests

- Authors' contributions

- Acknowledgements

- Funding statement

Because the schedule for publication is very tight, it is a condition of publication that you submit the revised version of your manuscript before 07-Dec-2018. Please note that the revision deadline will expire at 00.00am on this date. If you do not think you will be able to meet this date please let me know immediately.

Supplementary files will be published alongside the paper on the journal website and posted on

the online figshare repository (<https://figshare.com>). The heading and legend provided for each supplementary file during the submission process will be used to create the figshare page, so please ensure these are accurate and informative so that your files can be found in searches. Files on figshare will be made available approximately one week before the accompanying article so that the supplementary material can be attributed a unique DOI.

Please note that Royal Society Open Science charge article processing charges for all new submissions that are accepted for publication. Charges will also apply to papers transferred to Royal Society Open Science from other Royal Society Publishing journals, as well as papers submitted as part of our collaboration with the Royal Society of Chemistry (<http://rsos.royalsocietypublishing.org/chemistry>). If your manuscript is newly submitted and subsequently accepted for publication, you will be asked to pay the article processing charge, unless you request a waiver and this is approved by Royal Society Publishing. You can find out more about the charges at <http://rsos.royalsocietypublishing.org/page/charges>. Should you have any queries, please contact openscience@royalsociety.org.

on behalf of Prof Jon Blundy (Subject Editor)
openscience@royalsociety.org

Reviewer comments to Author:
Reviewer: 1

Comments to the Author(s)
Review of the article "Microplastic and synthetic particles ingested by deep-sea amphipods in six of the deepest marine ecosystems on Earth"
Reviewer: 1

The authors did a good job in amending the ms and provided a convincing response to all queries, except for the line 113 where authors declared the impossibility to test how fast the digestion is in the different species. I think there are some data available and the authors should make a further effort in this direction. The other minor issues have been answered in a comprehensive manner. Overall, however, I'm happy with the present version and I recommend its publication after this minor change.

Author's Response to Decision Letter for (RSOS-180667.R1)

See Appendix D.

Decision letter (RSOS-180667.R2)

22-Jan-2019

Dear Dr Jamieson,

I am pleased to inform you that your manuscript entitled "Microplastics and synthetic particles ingested by deep-sea amphipods in six of the deepest marine ecosystems on Earth" is now accepted for publication in Royal Society Open Science.

on behalf of Prof. Jon Blundy (Subject Editor)
openscience@royalsociety.org

Appendix A

Reviewer: 1

1. the study refers to "microplastic", but lacks of analytical (chemical or spectroscopic) analysis, which is absolutely necessary to confirm that the filaments they found are made of plastic and not of other materials (as often the case when made of cellulose or rayon or viscose). The fibres and fragments were indeed identified in terms of colour and shape from the material extracted from the animal.

RESPONSE: We very much take the point here of reviewer #1 (and #2) and so we took a sub sample of the fibres to a state of the art FTIR analysis facility at Shimadzu UK Ltd in Milton Keynes. We now have established that the fibres are a mix of synthetic polymer, semi-synthetic fibres and a few natural fibres, mostly mused in textiles etc. We have therefore been careful with the use of the word 'microplastic' as this does not always cover the range of fibres found, thus we have changed throughout, where appropriate, the terminologies of microplastic and microfibre, regardless the point remains that these were all found to be anthropogenic following the expertise analysis requested by the reviewers.

In the MS we have added:

In the Methods:

A subsample of fibres (n=15) spanning all trenches were analysed by Fourier-Transform Infra-red Spectrophotometer (FTIR; IR Tracer-100, Shimadzu, Japan) connected to an automatic infrared microscope (AIM-9000, Shimadzu, Japan) at the Shimadzu UK Ltd Laboratory Facility in Milton Keynes. Individual fibres were manually removed and transferred to the surface of FTIR reflective slides (Kevley Technologies, Ohio) (which provide a suitable background for reflectance) or transferred to a Specac DC3 Diamond Cell and compressed for transmission measurements (with background scans being taken through the diamond adjacent to the sample). The fibre was observed using the wide field camera to identify possible locations for further investigation, and the measurements were made in transmittance or reflectance mode (50 scans over approx. 20 s) using the Wide-Band MCT (mercury cadmium telluride) detector. For each fibre, three points were scanned and the results were compared to those in the Shimadzu materials library for matches or closest similarity. Some of the fibres which showed unusual structure were scanned in several places to reveal more about their chemical composition.

In the results:

Six of the 15 items analysed using FTIR were semi-synthetic cellulosic fibres, rayon and lyocell, the natural fibre ramie that are used in products such as textiles. Rayon is also used on absorbent hygiene products, and ramie is used in sewing threads, packing materials, fishing nets and filters. The rest included synthetic polymers such as Nylon, polyethylene (PE), polyamide (PA), or unidentified polyvinyls closely resembling polyvinyl alcohol (PVAL) and with most including an inorganic filler material. One fibre found in the Peru-Chile Trench at 7050m was clearly a polyethylene coated strand of polyester, possible representing fishing line. None of the 15 subsamples were found to be naturally occurring.

ALL FTIR data is shown in Table 2.

2. the manuscript needs to be carefully revised / supported for some definitions/assumptions as some of them could have important scientific implications :
- microplastic: the authors only specify the upper limit (≤ 5 mm)

RESPONSE: Added that the lower limit is 3333 μ m and cited Desforges, J.P.W., Galbraith, M., Dangerfield, N. and Ross, P.S., 2014. Widespread distribution of microplastics in subsurface seawater in the NE Pacific Ocean. *Marine pollution bulletin*, 79(1), pp.94-99.

- refs for the ingestion of microplastic by other marine organisms such as deep-sea fish and invertebrate (e.g., polychaetes)
- you take 3000 m as the maximum depth reached by the bathyal zone but this could be 4000 m depending on the definitions

RESPONSE: This really isn't the paper to redefine the world's biozones, some say 3000m and some say 4000m for the bathyal-abyssal boundary. We have therefore changed the sentence to read "...limited to depth 'within' the bathyal zone (<30000 m)..."

2. The results are not clearly illustrated. The histogram in Figure 4c is not clear and intuitive. The percentage relative to the amphipods is not represented and apparently comes from the blue. I would recommend to include an additional figure illustrating the different percentages of the different fibres and fragments to be consistent with the discussion.

RESPONSE: this exact graph has been added as Figure 4D

Introduction

Line 39. Please, add a reference

RESPONSE: Cózar, A., Echevarría, F., González-Gordillo, J. I., Irigoien, X., Úbeda, B., Hernández-León, S., Palma, Á. T., Navarro, S., García-de-Lomas, J., Ruiz, A. 2014 Plastic debris in the open ocean. *Proc. Natl. Acad. Sci. U. S. A.* 111, 10239-10244

Line 66, Please add a references

RESPONSE: Taylor, M. L., Gwinnett, C., Robinson, L. F., Woodall, L. C. 2016 Plastic microfibre ingestion by deep-sea organisms. *Sci. Rep.* 6, 33997.

Line 78. Please add a reference

RESPONSE: Ichino, M.C., Clark, M.R., Drazen, J.C., Jamieson, A., Jones, D.O., Martin, A.P., Rowden, A.A., Shank, T.M., Yancey, P.H. and Ruhl, H.A., 2015. The distribution of benthic biomass in hadal trenches: a modelling approach to investigate the effect of vertical and lateral organic matter transport to the seafloor. *Deep Sea Research Part I: Oceanographic Research Papers*, 100, pp.21-33.

Line 81. Please add a reference

Line 83. Please add a reference

RESPONSE: 81 and 83 are the same sentence, added:

Blankenship, L. E., Levin, L. A. 2007 Extreme food webs: Foraging strategies and diets of scavenging amphipods from the ocean's deepest 5 kilometers. *Limnol. Oceanogr.* 52, 1685-1697

Methods

Line 101. Add a reference here

RESPONSE: Lacey, N.C., Rowden, A.A., Clark, M.R., Kilgallen, N.M., Linley, T., Mayor, D.J. and Jamieson, A.J., 2016. Community structure and diversity of scavenging amphipods from bathyal to hadal depths in three South Pacific Trenches. *Deep Sea*

Research Part I: Oceanographic Research Papers, 111, pp.121-137.

Results

Line 146. Here you used "identified", but no specific analysis was made thus you cannot say that is microplastic.

RESPONSE: See opening response text above regarding inclusion of new FTIR analysis

Line 152. Please, add a graph

RESPONSE: There is already a graph showing these data – Figure 4C

Discussion

Line 183. Please add a reference

RESPONSE: Taylor, M. L., Gwinnett, C., Robinson, L. F., Woodall, L. C. 2016 Plastic microfibre ingestion by deep-sea organisms. *Sci. Rep.* 6, 33997.

Results

Line 214, Please add a reference that state that blue fibres are derived from fishing nets

RESPONSE: Turner, A. 2017 Trace elements in fragments of fishing net and other filamentous plastic litter from two beaches in SW England. *Environ. Pollut.*, 224, 722-728

Line 215. Please add a reference for shallower ecosystems

RESPONSE: Lusher, A., 2015. Microplastics in the marine environment: distribution, interactions and effects. In *Marine anthropogenic litter* (pp. 245-307). Springer International Publishing

Table and Figure

Table 1. Geographic coordinates should be provided in degrees Figure 2.

RESPONSE: Changed to DD in Table 1. Coordinate grid already included in Figure 1.

Please provide size range Figure 3. Again please provide the scale, and include in the legend of the magnification used.

RESPONSE: done

Reviewer: 2

Abstract

- Line 11. I think it is time to move away from GROWING. Microplastics are now recognized worldwide as a form of pollution.

RESPONSE: Changed to: Whilst there is now an established recognition....

Line 12: detrimental, only in the lab, in the main body text some more information on this could be added to support its inclusion in the abstract

RESPONSE: It is only proven categorically in the lab, but to accommodate this technicality we have changed the sentence to read: ... and the detrimental effects this may have on marine animals...

-Line 23: change negligence to debris

RESPONSE: Changed as suggested

Introduction

-Line 26: GROWING same as introduction

RESPONSE: Changed to "There is now an established appreciation..."

-Line 27: please address the references and update accordingly. [1-3] would be better replaced with a review to represent the breadth of the subject area and current knowledge. For example, GESAMP released a through review in 2016, which is an update and extension of Lusher 2015.

Reference 1-3 are now (as suggested)

RESPONSE: GESAMP (2015). "Sources, fate and effects of microplastics in the marine environment: a global assessment" (Kershaw, P. J., ed.). (IMO/FAO/UNESCO-IOC/UNIDO/WMO/IAEA/UN/UNEP/UNDP Joint Group of Experts on the Scientific Aspects of Marine Environmental Protection). Rep. Stud. GESAMP No. 90, 96 p

GESAMP (2016). "Sources, fate and effects of microplastics in the marine environment: part two of a global assessment" (Kershaw, P.J., and Rochman, C.M., eds). (IMO/FAO/UNESCO-IOC/UNIDO/WMO/IAEA/UN/UNEP/UNDP Joint Group of Experts on the Scientific Aspects of Marine Environmental Protection). Rep. Stud. GESAMP No. 93, 220 p.

Lusher, A., 2015. Microplastics in the marine environment: distribution, interactions and effects. In Marine anthropogenic litter (pp. 245-307). Springer International Publishing.

-Line 28: [4] is not a suitable reference, it is a fish paper. find a better reference

RESPONSE: While the statement here is not related to fish, we have deleted it as the source of the statement is actually from the next paper cited, so, reference deleted.

-Line 29 and 30: both references used here [5,6] need to be clearly stated that they are estimations based on limited empirical evidence and as such should be used with caution.

RESPONSE: The sentence already starts with "An estimated" so this is already clear....

-Line 31: remove this reference and include reference to the Anthropocene,

Zalasiewicz, J., Waters, C.N., Wolfe, A.P., Barnosky, A.D., Cearreta, A., Edgeworth, M., Ellis, E.C., Fairchild, I.J., Gradstein, F.M., Grinevald, J. and Haff, P., 2017. Making the case for a formal Anthropocene Epoch: an analysis of ongoing critiques. *Newsletters on Stratigraphy*, 50(2), pp.205-226.

RESPONSE: Changed as suggested

-Line 41: use a more recent reference. Woodall et al. might be suitable

RESPONSE: Deleted REF 8 as Woodall is number 9 anyway, and this does not change the context of the sentence.

-Line 41: the opportunities for dispersal are not the only reason plastics reach the deep sea, i would like to see other reasons for plastic accumulating in the deep sea, settling rates, density, biofouling etc. Some of the recent publications highlight and discuss these methods.

RESPONSE: This comment does not really make sense. We state the opportunities are

limited (with citation), the rate that it settles and biofouling (which largely doesn't exist at these depths) do not offer a tangible alternative, plastic sink to the deep sea and at that point there is nowhere else to go. That is our point.

-Line 52: change reference [17] this is a terrestrial and aquatic study and not relevant to the deep sea

RESPONSE: Changed to Bakir et al., 2014

-Line 53: curious to why the word RAIN is used here, if the discussion of MPs route to the deep sea is expanded it may be relevant but at the moment it seems out of place

RESPONSE: 'Rain' changed to 'input'

-Line 58: reference [56] is a methods paper i would recommend including a study such as Paul-Pont et al. 2016.

RESPONSE: Agreed, changed to Paul-Pont, I., Lacroix, C., Fernández, C.G., Hégaret, H., Lambert, C., Le Goïc, N., Frère, L., Cassone, A.L., Sussarellu, R., Fabioux, C. and Guyomarch, J., 2016. Exposure of marine mussels *Mytilus* spp. to polystyrene microplastics: Toxicity and influence on fluoranthene bioaccumulation. *Environmental Pollution*, 216, pp.724-737.

is reference [27] accepted? if not please include a peer-reviewed publication.

RESPONSE: It is now and changed to: Courtene-Jones, W., Quinn, B., Gary, S.F., Mogg, A.O. and Narayanaswamy, B.E., 2017. Microplastic pollution identified in deep-sea water and ingested by benthic invertebrates in the Rockall Trough, North Atlantic Ocean. *Environmental Pollution*, 231, pp.271-280.

-Line 62: reference [31] should be updated to be a MP study rather than general plastic debris, suggest GESAMP 2016

RESPONSE: Changed to GESAMP as suggested

-Line 65; reference [37] should be replaced with a pelagic study, not on fish, suggest Lusher et al. 2014 or Enders et al. 2016 both are from the Atlantic, and Enders is middle of the ocean basin.

RESPONSE: Changed to:

Lusher, A.L., Burke, A., O'Connor, I. and Officer, R., 2014. Microplastic pollution in the Northeast Atlantic Ocean: validated and opportunistic sampling. *Marine pollution bulletin*, 88(1), pp.325-333.

Reference [38] should be replaced with a more up to date reference, there are many recent sediment papers.

RESPONSE: Changed to:

Galloway, T.S., Cole, M. and Lewis, C., 2017. Interactions of microplastic debris throughout the marine ecosystem. *Nature Ecology & Evolution*, 1, p.0116.

[40] is a macropalstic paper and suggest it is omitted from this list

RESPONSE: Reference deleted

Methods

-The methods used are sound as written but present a significant flaw in this research. There is no confirmation of the visually identified plastics. As it stands this study presents POTENTIAL microplastics. As recent research has highlighted, visual identification is not recommended on its own, and it needs to be supported by further techniques. I urged the authors to address this significant error. I am highly opposed to publishing anything that does not even attempt to confirm the authors "visual "

identification skills. A subset 10% of potential particles would improve interpretation of visual analysis greatly.

- I would like to some annotations to the images of the amphipods to show the location of the hindgut

Results

Line 146- remove individual

RESPONSE: Done

Line 148- remove sampled

RESPONSE: Done

Line 149- remove sampled

RESPONSE: Done

Discussion

Line 170: reference [38] is a very old study, please update with recent advances in the literature

RESPONSE: Changed to Desforges, J.P.W., Galbraith, M., Dangerfield, N. and Ross, P.S., 2014. Widespread distribution of microplastics in subsurface seawater in the NE Pacific Ocean. Marine pollution bulletin, 79(1), pp.94-99.

Line 168-181- i would like to see some discussion on potential for airbourne contamination, this section would also be greatly supported by inclusion of chemical analysis of potential microplastics. For example, If Nylon is identified, then the authors could infer possible sources.

RESPONSE: We believe this to be beyond the scope of the study. This study is identifying the fact that the fibres exist within organisms at these depths. Any detailing of how they got there beyond 'sinking' would be pure conjecture. We believe that a larger follow-on study looking into sources would indeed be very interesting but the real mechanism, sources, timescales involved are really beyond the scope of this initial study.

Line 186: i am not convinced that POPs and associated chemicals to plastics are relevant to this manuscript as the authors do not study the effects, its inclusion would be relevant if better introduced. I feel that more discussion on sources and routes of MPs to the deep sea sediment is more important.

RESPONSE: see comment above. Transport and mechanisms are pure speculation but it is actually known that POPs aggregate to plastics and thus the presence of plastics with increase POP contamination, which, as explained in the paper, correlates to the findings of Jamieson et al 2017, Nature Ecol. Evol.

Line 198- It is important to note that these studies are from laboratory exposure not the marine environment

RESPONSE: Changed environment to organisms.

Line 219 replace negligence with debris

RESPONSE: Done

Figures

Figure2. if possible i would love to see an indication of where the hind gut is Figure3.

RESPONSE: added as indicated by *

please include a typical fragment –

RESPONSE: these make terrible images that we don't feel offers the reader anything, they just look like 'blobs'. Illustrating the difference between a 2mm diameter fragment and a 2mm fibre seems to detract from the seriousness of the paper, and it is hard to visualise such tiny objects in a meaningful way

Figure 4. would be interesting if a) was included on the map of the locations.

could consider pie charts for presence/absence percentage

RESPONSE: As the locations are so far and wide, adding on extra data makes the map harder to read, in our opinion.

c) minor error in the key., Black Fibre needs F to be f

RESPONSE - fixed

Appendix B

Reviewer comments to Author:

Reviewer: 2

Comments to the Author(s)

Review: RSOS-172333

Microfibre ingestion in size of the deepest marine ecosystems on earth

Dear Authors,

Thank you for producing this manuscript. You have investigated the presence of microfibres/particles ingestion by amphipods in the deep sea. This information is timely and provides a new insight into deep sea contamination. The introduction is well references, although some references can be updated. The object of the study is clearly stated. I am concerned about the number of individuals per site, although I do understand the difficulties in collecting specimens from such depth. I would urge the authors to therefore reduce the speculation of site differences in the results due to the low number of individuals. There are two few specimens per sites to carry out sufficient statistical analysis. I lack size descriptions of particles identified. Please include more discussion on other papers looking at deep sea organisms. The discussion has a large text on toxicological effects, I feel too much weight is given here, considering the study does not focus on toxicity. I would like to see more information on the effect of KOH, and how the sites are not comparable. Please expand.

Response:

The reviewer has highlighted a number of points of concern. We address some of the reviewers concerns below and within the specific comments listed.

1. Number of individual per site and statistical analysis.

It is not clear if the statements are relating to the lack of statistical analysis and perceived low sample size are criticisms of the manuscript which the reviewer wishes us to address. There is no guidance regarding what the reviewer feels is an appropriate sample size or how an appropriate sample size might be calculated. This makes the comment very difficult to address. In many cases just increasing the sample size of study does not provide any benefit. If we took 10 samples and found no particles then there is a justification to ask for an increased sample size because if they were in very low levels within the population we may have missed them. However, we have found microparticles in 50% to 100% of the individuals (depending on the trench) subsampled at random from a larger population. A subsample of these were subjected to FTIR and were identified as microplastics. When

the question is focused around whether something is essentially present or absent and it is found to be present, there is no need to increase the sample size.

The reviewer has highlighted that the comparisons among the sites is a potential problem with the manuscript but we have merely presented the results as means, standard errors and percentages and described what we have found, which is standard practice. The only statistical analysis that has been carried out is the relationship between microparticles and depth in the Kermadec Trench. 10 individuals for a statistical test based on means or ranks is an ample sample size depending on the effect size you are looking to detect. Given that the samples were chosen randomly from a greater number of individuals collected from the site, the prevalence and mean items were the same at all depths and the relationship was not significant, there is no need to increase the sample size.

2. Comparability of the sites

We have not undertaken statistical analysis comparing the number of microparticles among the trenches. This is because we feel time is a confounding factor in the analysis. We cannot say whether differences among trenches are a real phenomenon related to some mechanism that results in plastics being transported to the deep sea because differences may be purely related to accumulation over time. The trenches have been sampled between 2008 and 2017. We have no temporal data in order to understand accumulation rates and adjust for it in an analysis. However, these data serve as a valuable baseline assessment from which we can understand future microplastic ingestion in amphipods in hadal trenches and whether it is increasing.

3. Include more discussion on other studies looking at deep-sea organisms

There are very few studies that look at the ingestion of the microplastics in deep-sea organisms. In doing a Web of Knowledge search using the search string "microplastic*" AND "ingest*" AND "deep sea", only 8 papers appear. Of these only two examine whether microplastics have been ingested by deep-sea organisms deeper than 1000 m. These are Taylor et al 2016 (10.1038/srep33997) and Courtene-Jones et al. 2017 (10.1016/j.envpol.2017.08.026). There are two other papers which examine vertically migrating fish by Choy et al 2013 (10.3354/meps10342) and Lusher et al 2015 (10.1093/icesjms/fsv241) but these papers only go to a depth of 500m. Lusher et al. (2015) only used visual methods to identify microplastics. There really is very little information out there on this topic to

discuss. We have added some additional text in the discussion which includes these studies in a bit more detail.

4. Discussion on toxicological effects.

We have removed this from the discussion.

5. Effects of potassium hydroxide (KOH)

This paper is not designed to look at the effects of KOH on sample preparation. We do not have any data in order to discuss it. We have cited a recent paper, which examined whether KOH was an appropriate solution to dissolve organic material for microplastic research and they concluded that using a 10% solution of KOH was an appropriate method. We have commented further on this in relation to one of the “specific comments”.

Specific comments (line numbers supplied by authors)

Title: update to be microplastics or microparticles to truly reflect the results

Response: We have changed the title to “Microplastic and synthetic particles” as this reflects both the plastic ingestion and the man-made fibres that were ingested by the amphipods.

Introduction

Line 30: please update this value, 322 was the latest value from Plastic Europe in 2015.

Response: We were unable to find this value on Plastic Europe website. In light of this, we will stick with the value in Erickson et al. 2014 (10.1371/journal.pone.0111913) rather than placing in an unreferenced source.

Line 30: litters is a very colloquial term, I would suggest the authors state occurs/is present.

Response: We have altered the sentence to “... of plastic is now present in the oceans,...”

Line 57: by definition fibres are not primary microplastics, they are produced through the breakdown. I would also advise the authors to avoid terminology such as primary and secondary microplastics. There are three types of microplastics. Those that are produced for use in the microscale, those that breakdown during use, and those that breakdown after use/once they are environmental contaminants.

Response: We have updated lines 55 to 58 and “primary” and “secondary” has been removed.

Line 57: reference 18 is not needed here.

Response: Reference 18 has been removed.

Methods

Line 107: 10 individuals per site is very low for replicates. I would therefore highlight that any differences between sites cannot be classed as significant. Further replicates are needed.

Response: We have not undertaken a statistical test to compare among locations because the analysis would be confounded by time. Therefore, we have not made any inference about whether locations are significantly different in the statistical sense. We took 10 individuals per site and found that between 50% and 100% of them contained microparticles. FTIR identified some of these as plastic. Therefore, we have answered the objective of the paper. There is no discussion about why one area has higher instances of ingestion and no statistical analysis comparing the trenches so the reviewers comment is not applicable.

Line 108: remove baited. (is repeated in 109, so not needed twice)

Response: "baited" removed from line 108

Line 110: Change to ... could affect "further study."

Response: We have removed "such as this study."

Line 113: Question: do the authors know how fast digestion is in this species, could digestion happen during capture? I would like to see some discussion around this point.

Response: We do not know how fast digestion is in these species. Feeding studies are extremely difficult in hadal species because the amphipods arrive dead on deck as a result of decompression mortality.

Line 118: Question: Are the authors aware of any previous research that shows the % of ethanol that affects plastic?

Response: We have already cited Courtene-Jones et al (2017, doi 10.1039/C6AY02343F) in the next line which states that ethanol does not appear to significantly impact or degrade microplastics.

Line 141: why was only 15 particles subjected to FTIR? Does this follow MSFD and NOAA that recommend 10% of overall subsample based on size?

Response: There were 122 microparticles observed in the study and we subjected 15 of them to FTIR. This means we analysed 12% of the particles

found on the filter papers, which exceeded the request of a reviewer for the previous submission and are in line with the MSFD guidelines. The FTIR has shown that some of these items ingested by the hadal amphipods are microplastics, which is the salient point of the manuscript. This is the first record of microplastic ingestion in the hadal environment.

Line 149: Why were both transmittance and reflectance used? They can provide different results, ATR is better at distinguishing between semi-cellulosic particle.

Response: The fibres were analysed by transmittance and reflectance but we found that transmittance gave the most reliable results. We've corrected the methods section to say that all fibres were analysed by transmittance to avoid any confusion. ATR was not used as the samples were of variable morphological characteristics and in most cases unsuitable for ATR as direct contact can deform or move the sample. The samples would have had to be transferred to a hard crystal for measurement and transmission gives better results in most cases when using the AIM-9000. The AIM-9000 gives very high quality results in transmission and reflective modes, which would have previously needed ATR.

Results

Line 155: fibres AND fragments

Response: The text has been altered to "fibres and fragments".

Line 165: 122 items in total. Were all particles plastics/semi-synthetic, how many were rejected based on the FTIR results?

Response: As stated in the methods, there were 15 items out of 122 analysed by FTIR chosen at random from a subset of amphipods that contained the various types of microparticles. The analytical summary and correspondence provided by the laboratory indicated that they were happy with the identifications of the 15 particles analysed.

Line 176: ramie, correct spelling please

Response: ramie is spelt correctly on line 176.

Line 181: Please discuss the potential effect of KOH on particles.

Response: It does not seem appropriate to discuss the potential effect of KOH on particles in the results section. The study did not look at the effects of KOH on the different types of plastics. There are no results to report and therefore discuss. We have placed the following sentence in the methods

section which supports the use of KOH along with an accompanying reference:

“KOH has been shown to be a suitable solution to dissolve the guts of marine fauna, leaving the majority of microplastics unaffected (Kühn et al 2017; [10.1016/j.marpolbul.2016.11.034](https://doi.org/10.1016/j.marpolbul.2016.11.034))”

Discussion

Line 188: please include other papers on deep water organisms. Taylor etc.

Response: The statement is making a comparison to coastal water habitats not other deep-sea habitats. Therefore, it is not appropriate to cite Taylor et al 2016 (10.1038/srep33997) with this statement. Reference to this paper and other papers focusing on deep-sea habitats are made in appropriate places throughout the text.

Line 193-195: Due to the date differences, are the samples comparable? I lack discussion here.

Response: The differences in sampling dates mean that time is a confounding factor in the analysis. We have analysed whether there are differences in microparticle numbers ingested among depths in the Kermadec Trench as these samples were taken within a short time window. It is difficult to draw any conclusions about why amphipods in one trench have higher numbers of ingested microparticles when there is a 9 year difference in sampling. The difference could be the result of greater time in which plastics have accumulated in the area rather than any differences in the way plastics accumulate in the area or the mechanism by which plastics reach the seafloor. We have added some text to the discussion related this point.

Line 214-218: this study does not show these. I suggest the authors make it clear

Response: We have removed the section on potential links between plastic and persistent organic pollutants.

Conclusion

Line 233: here you have used world's whereas earlier in the document you have used World's. Please be consistent.

Response: This was a typing error. We have changed the all of the “world's” to “Earth's” so it consistent with the title.

Line 235: which factors? Please include,. Would like to see these factors discussed in more details in the discussion.

Response: We have added some additional text to the discussion which states some of the factors by which plastics can enter the deep sea. These appear between lines 210 and 214 of the new submission. We have removed the statement relating to the higher ingestion of microplastics in the Marina Trench.

Line 236-237 should be removed as no a key result from this study

Response: We have removed part of the sentence that relates to toxicological effects.

Line 238: This study does not show plastics are culminating and accumulating. Should not be a conclusion.

Response: We acknowledge there is no temporal aspect to these data so therefore we cannot categorically say that plastics are “accumulating” in the trenches we have studied. We do feel that we are justified to say plastics are “culminating” because culminate means to reach a final point. Hadal trenches are the deepest point of the ocean and microplastics arriving on the seafloor of the hadal trenches are at culmination of a journey that started from the surface waters or further afield. The bottom of trenches are the final point at which they are bioavailable unless they float back to the surface. We have removed “accumulating” replaced it with “ingested, culminating and are therefore bioavailable...”. This reflects the results and indicates that plastics are ingested by organisms within the deepest ecosystem on the planet.

Reviewer: 3

Comments to the Author(s)

The manuscript presents observations of microplastics from amphipods sampled from hadal depths. It covers six locations across the general Pacific region. From the observations authors conclude ‘it is likely that all marine ecosystems ... have been impacted by anthropogenic debris’. I suggest that the authors did not require their new data to make this conclusion, therefore need to reconsider what their data bring to the crowded field of microplastics research.

Response:

We have rephrased the conclusions of the paper so that they highlight that this is the first occurrence in the hadal zone and trench habitat of microplastics being ingested by deep-sea organisms and the deepest record

to date. The FTIR analysis indicates that microplastic and synthetic fibres are being ingested by organisms living in some of the deepest places on Earth. This is a first discovery and adds yet another marine habitat, in this case deep sea trenches, to the list where organisms are ingesting man-made fibres. This discovery is the first recorded incidence. It is not an isolated occurrence given the geographical area we have covered. This vastly increases the previous maximum depth that microplastics have been confirmed as being present in the guts of deep-sea organisms.

There are only two other papers that we are aware of that examine the ingestion of microplastics in deep-sea organisms past 1000m. These are Taylor et al 2016 (10.1038/srep33997) and Courtene-Jones et al (2017, doi 10.1039/C6AY02343F). Woodall et al 2014 (10.1098/rsos.140317) only examines whether microplastics adhered to external structures of corals sampled around 1000m. There are only 6 individuals in the Taylor et al 2016 (10.1038/srep33997) paper that have microfibers associated with them, 2 of which were crustaceans and only one of which 1 had confirmed microfibers in their stomach. These fibres were identified using a polarised light source. There is no mention of FTIR analysis in the paper. Courtene-Jones et al (2017, doi 10.1039/C6AY02343F) provide data on 3 deep-sea species with sample sizes ranging between 7 and 49 from the Rockall Trough. They use FTIR within their study but do not look at crustaceans as we have done.

In relation to what has been previous published on microplastic ingestion within deep-sea organisms including species that ingest microplastics, depth of occurrence, methodological technique used and geographical extent, this paper clearly has a place in the “crowded field of microplastics”.

Specific comments follow

Abstract

Line 13-14 untrue- Ubiquity has been shown across geography and depths. Agreed not the depths presented in this paper, how everywhere microplastics have been looked for they have been found. Suggest rephrasing this statement.

Response: We have altered these lines to be specifically related to depth and ingestion which is the primary aim of the paper. It now reads: “the ocean depth at which such contamination is ingested by organisms has still not been established.”

Line 22 Plastic polymer names should be given in full instead of the abbreviations

Response: We have changed the abbreviations to full names as requested.

Line 25-26, this is not new and the data in this paper are not required for this hypothesis- see main comment

Response: We have altered the text so that it reads:

“indicating that anthropogenic debris is bioavailable to organisms at some of the deepest locations in the Earth’s oceans.” rather than “indicating that no marine ecosystems that have not been impacted by anthropogenic debris.”

The altered text now clearly states that the microplastics are bioavailable in the deepest locations within Earth’s oceans, which highlights both the novelty of the results and the new data within this manuscript which is the deepest possible record.

Keywords Spelling ‘microfiber’

Response: We have changed the spelling to “microfibre”.

Introduction

Line 36-41- Suggest removing, not relevant to the paper

Response: We have removed some of the text within these lines as the reviewer suggested but kept other parts because we feel they are required to frame the question relating to at what depths are microplastics bioavailable to marine organisms. We have removed the following sentence: “This is evidence from the proliferation of studies published on plastics, with a basic Google Scholar search of ‘marine microplastics’ showing an exponential growth of scientific papers from 15 in 2006 to 997 in 2016. Within such data there is an intuitive and expected bias towards.” Then we have altered the following sentence to: “The investigation of microplastic ingestion by marine organisms has largely focused on shallow water habitats given the ease of sampling these locations yet we know very little about their ingestion in the deep sea.” The edited sentence highlights the current bias in the microplastic literature towards shallow water habitats. This now sets up our primary research question of whether microplastic pollution extends to the full depth of the ocean and if it bioavailable.

hLine 53- See Koelman et al 2016 and Bakir et al 2016 for discussion on this. It is not as certain as authors suggest. It is important to qualify this statement such as ‘Plastic could act as vectors...’

Koelmans, A. A., Bakir, A., Burton, G. A., & Janssen, C. R. (2016). Microplastic as a vector for chemicals in the aquatic environment: critical review and

model-supported reinterpretation of empirical studies. *Environmental science & technology*, 50(7), 3315-3326.

Bakir, A., O'Connor, I. A., Rowland, S. J., Hendriks, A. J., & Thompson, R. C. (2016). Relative importance of microplastics as a pathway for the transfer of hydrophobic organic chemicals to marine life. *Environmental pollution*, 219, 56-65.

Response: We have condensed the section on the link between plastic and persistent organic pollutants in the introduction. We now have a single line in the introduction which states:

“There is also the potential for plastics to act as a vector for pollutants including persistent organic pollutants (e.g. polychlorinated biphenyls)....” . We have used the references suggested by the reviewer to support the statement.

Line 55 ref 16 is a review and does not support this statement in fact it states ‘Hydrophobic contaminants are enriched on microplastics, but the available experimental results and modelling approaches indicate that the transfer of sorbed pollutants by microplastics is not likely to contribute significantly to bioaccumulation of these pollutants.’

Response: We have removed this line from the introduction.

Line 63-Might be useful to add in this sentence that the negative impacts documented to date are those from physical mechanisms of the microplastics, as these are the impacts listed.

Response: We have not altered the sentence because we do not feel that adding “physical” somewhere would add to the sentence.

Line 72- Fischer et al already show microplastics in hadal sediments. Why is it not appropriate to extrapolate plastic presence to greater depth and organism ingestion to those inhabiting those depths? The mechanisms for microplastic sinking given in the papers cited, continue through the depths of the ocean.

Response: On line 72 we do not state that it is not appropriate to extrapolate that microplastics have not reached the deepest parts of the ocean. We have merely stated that we do not know if they are present at deeper locations than have been sampled because there is no published data to show that they have. Fischer et al 2015 (10.1016/j.dsr2.2014.08.012) collected microplastic from box corers on the upper margins of the Kuril-Kamchatka Trench and adjacent abyssal plain. The deepest sampling point was 5768m for the sediments, which are not hadal depths and are not the deepest point of the Kuril-Kamchatka Trench

(~10,500m). The paper also did not look for microplastic ingestion by the organisms which were collected during the research cruise. If microplastics are present in the environment you can hypothesise that it is likely they are being consumed by the organisms living there but finding the particles in the organisms' guts is what is required in order to categorically say they are being ingested. We have taken that step within this study and have provided the first evidence that microplastics are ingested by organisms living in hadal environments. Until you have confirmation that microplastics are present, then any extrapolation to deeper depths and ingestion by those organisms living there is still a hypothesis.

Methods

Line 118- Was the container cleaned?

Response: The container in which the samples were preserved in was not cleaned. However, the animals arrived on the deck dead as a result of decompression mortality. We did not examine any external structures like gills or maxillipeds in case microplastics had adhered to the body during ascent to the surface or were already in the container. The amphipods were dead when they were preserved in ethanol and therefore unable to feed on any particles that might have been in the container. It is highly unlikely that a piece of plastic can passively enter the amphipods mouth, pass through the foregut and enter the hindgut during preservation.

Line 123- What was lab coat made from? Where there any other controls about clothing worn in the lab and during sample collection?

Response: There were no controls in place during sample collection because the original collection was not designed specifically for microplastic studies. There were controls in place to stop the amphipods feeding on the bait. This means that if microplastics were present in the stomachs of the mackerel they would not be transferred to the individuals collected. Given the previous point whereby we highlight that the animals were dead on arrival on deck, it is highly unlikely that the microplastics would have contaminated by clothing during their transfer to the preservation jar. The laboratory coat was a polyester-cotton mix and was white. We did not find any white fibres in our samples nor on the control filter papers we placed inside the fume hood. No additional clothing controls were made during the laboratory work. We have added the following line to the methods section: "We did not find any white fibres that may have been contamination from the white laboratory coat worn during sample preparation."

Results

Line 175-181 Suggest deleting. Not sure in the value of suggesting which products the microplastics could have come from. Polymers are used over a vast range of products and it is not possible to track microplastics to a source. Maybe a simple statement that says a wide varied of polymers were observed and therefore they likely come from a wide variety of products.

Response: We have removed the sentences regarding the products that these plastics may have originated from.

Discussion

An important point for the discussion is that light microscope screening into categories does not work, as stated lines 172-173.

Response: We have added the following text to the discussion:

“However, it is clear from the FTIR analysis and previous works that the colour-based categorisation is not an adequate method to identify whether a microparticle is indeed of plastic origin (Gago *et al.* 2016; 10.3389/fmars.2016.00219). The range of plastic found in the hindguts of the amphipods included PE, PA, and polyvinyls resembling PVAL or PVC but we also found other synthetic polymers that are not plastics (e.g. ramie, lyocell).”

Line 196- Suggest deleting as the crude sorting process was not confirmed with FTIR. Therefore discussion results based on a classification they disprove is not useful. In addition I am unsure how useful it is to compare between studies just using plastic colour as they could have been of any material seen or unseen in the new data set.

Response: We have kept the crude colour-based categorisation because we felt it helps place the spectrum of samples in context with what has been found in Pacific surface waters. Many publications still do provide the crude colour-based categorisation. Otherwise the alternative is to undertake FTIR on every sample collected regardless of how many samples are found within the stomachs of the organisms collected.

Line 209 - See comments associated with line 53. You really need to state that most lab studies use concentrations and models that are not representative in the wild see review by Galloway et al (2017) for discussion.

Galloway, T. S., Cole, M., & Lewis, C. (2017). Interactions of microplastic debris throughout the marine ecosystem. *Nature ecology & evolution*, 1(5), 0116.

Response: We have removed line 53 and added the reference by Galloway et al. 2017 (10.1038/s41559-017-0116) in appropriate places within the text.

Line 222 Trophic transfer conclusions are based on lab studies using very high concentrations. Therefore the discussion needs to reflect this.

Response: We have added the following sentence to highlight that any inference on trophic transfer has been made based on experiments: "These studies were conducted under experimental conditions using high concentrations of microplastics but their results indicate the possibility of microplastics transferring among individuals (Farrell et al., 2013; Setälä et al., 2014)."

Conclusion

Line 235- Due to limited sampling not sure it is an appropriate to conclude higher ingestion rate in the Mariana Trench.

Response: We have removed this sentence.

Line 236 Suggest amending this sentence. Physical impacts have been shown not chemical ones.

Ref 39 does not support this statement.

Response: We have removed part of the sentence that relates to toxicological effects and made sure the references are correct.

Line 242 This is not a new conclusion see ref 7 cited in this manuscript

References

Response: Woodall et al. 2014 (10.1098/rsos.140317) examined microplastics in sediments from a series of deep-sea habitats but did not study a trench habitat nor went down to ~10,500m. The majority of their samples were around 1000m depth with a single data point at 3500m within the Mediterranean Sea. They did not examine whether the material had been ingested by the organisms living at these depths but only whether they adhered to the surface of corals in 4 specimens ranging in depth from 500 to 800m. This study is the first to report that microplastics have been ingested by organisms living at hadal depths and within trench ecosystems. The conclusion is still valid.

Line 295, ref 8 spelling 'the'

Response: We have corrected the spelling.

Line 394 ref 48 Italics for scientific name

Response: We have put the species name in italics.

Line 403 ref 52 check capitalisation

Response: Reference corrected

Figure 3 legend- Suggest change '3' to 'three'

Response: There is no "3" in the Figure 3 legend.

Table 1- Give details on gear OBS and Latis

Response: We have provided a reference and full names of the gear used to collect the amphipods in Table 1's legend.

Appendix C

Royal Society Open Science Editorial Office
Publishing Section
6-9 Carlton House Terrace
London
SW1Y 5AG

School of Natural and Environmental Sciences
Ridley Building
Newcastle University
NE1 7RU

Dear Editorial Board,

Manuscript revision (RSOS-180667)

We have revised the following manuscript: "*Microplastics and synthetic particles ingested by deep-sea amphipods in six of the deepest marine ecosystems on Earth*".

We thank the reviewers for their comments and address them in red below.

Yours faithfully,

Dr Alan Jamieson

Response to reviewers

Reviewer comments to Author:

Reviewer: 2

Comments to the Author(s)

Thank you for addressing my previous concerns. I have a few more minor comments

In your response to: Introduction Line 30: please update this value, 322 was the latest value from Plastic Europe in 2015.

You stated that: We were unable to find this value on Plastic Europe website. In light of this, we will stick with the value in Erickson et al. 2014 (10.1371/journal.pone.0111913) rather than placing in an unreferenced source.

Using data from three years ago is inappropriate. Here is the reference. See page 16. All scientific research has presented this data from earlier years, I see no reason why the authors cannot/refuse to use this data

<https://www.plasticseurope.org/en/resources/publications/274-plastics-facts-2017>

We thank the reviewer for providing the information required and have now included this value in the manuscript with an appropriate reference at the end.

I reiterate that 10 individuals per site is not an appropriate number, there is reference in the literature that 20-50 individuals are required to give a good description of contamination loads. Especially in areas as remote as the trenches. I highly urge the authors to increase the numbers of individuals at each site, especially when they say there are more individuals to use for analysis. This will have a better impact on the research community and will not be looked at as "another" presence and absence paper.

We did not say that there were more individuals that could be used for analysis. We stated that the samples for this study were chosen randomly from a larger pool of samples. Since the work was carried out nearly two years ago, a large proportion of the material has been used or is in the processes of being used for other research; some of which requires non-destructive sampling. Given how precious, unique and difficult these samples are to obtain and with no current plans to return to these areas, we cannot justify and do not have the capacity to do these additional samples. The manuscript addresses whether microplastics are ingested by hadal fauna, which was the primary aim of the work. These results still remain the first records of hadal fauna ingesting microplastics as well as the deepest records. Depending on how you define "impact", we feel this work will have an "impact" at scientific and societal levels. We also share the underlying opinion about just "another presence/absence" paper, like the reviewer states, that was in part the motivation for doing this study using extraordinary rare samples from the maximum depths of the ocean – to prove that plastic ingestion is occurring everywhere and that ever deeper presence/absence papers are not being all that helpful. This paper should bookend all these studies by showing it extends to full ocean depth and as a result should be highly cited in that context.

Line 113: Question: do the authors know how fast digestion is in this species, could digestion happen during capture? I would like to see some discussion around this point.

Response: We do not know how fast digestion is in these species. Feeding studies are extremely difficult in hadal species because the amphipods arrive dead on deck as a result of decompression mortality.

I thank the authors for this response. Such discussion would be useful in the document. Digestion times are highlighted in coastal studies, and even staytong the ansers are not clear but furture research would be needed would be suffice for the discussion.

We have added the following sentences to the end of the discussion that addresses the reviewers request:

"The extent to which deep-sea amphipods can disperse microplastics across the seafloor is currently unclear. This is because their digestion and defecation rates are currently unknown."

Reviewer: 1

Comments to the Author(s)

The authors generally did a good job in amending the ms and provided a convincing response to all queries, as detailed here below:

1. Number of individual per site and statistical analysis: The authors justified the reason for the reduced number of samples. The study focuses more on the presence or absence of microplastic in organisms. FT-IR analysis have been added clarified in the text.

No response required.

2. Comparability of the sites: the authors have answered in a comprehensive manner, limiting themselves to highlighting the individual results in the different sites, specifying that they do not know what the phenomena responsible for these different accumulations could be.

No response required.

3. Include more discussion on other studies looking at deep-sea organisms: the authors did not include Choy et al. 2013

This section of the discussion focuses on benthic invertebrates. Choy et al. 2013 paper focuses on pelagic fish up to depth so 400m. The Choy et al. 2013 does appear in the discussion later on discussing the mechanisms by which microplastics can be transferred to the deep sea. It is reference 46.

4. Discussion on toxicological effects: this part of the discussion has been omitted in the amended removed

No response required.

5. Effects of potassium hydroxide (KOH): the authors have have included more detailed information on the treatment with KOH and added a reference that explains more about the treatment

No response required.

Specific comments: all the comments have been taken into consideration by the authors, except for the line 113 where authors express the impossibility to test how fast the digestion is in the different species.

Please see the response to reviewer 2 regarding digestion rates.

I'm happy with the present version and I recommend its publication as it is

Appendix D

Dear Editor

As per an off line email conversation, the paper has been accepted subject to one last cryptic comment by reviewer 1. They were asked to clarify this statement on November 28th, it is now January 18th.

This paper was submitted in the autumn of 2017, the first round of reviews led us to go on and do another set of analyses that greatly enhanced the paper and it was resubmitted in December 2017. Since then it has taken THIRTEEN months going through just a few rounds of reviews of which I don't recall any of the comments really improving the manuscript at all. It is becoming very frustrating and we have just lost the last two months waiting for clarification on another cryptic comment, that regardless of the outcome of the query (if known) wouldn't change anything in the paper anyway.

I acknowledge that you have tried to contact the reviewer again and that you are holding the resubmission open until the end of January, but my colleagues and I are going to Antarctica on Monday and therefore I am going to resubmit exactly the same version that we submitted months ago in light of no further info from the reviewer. After much deliberation, we have decided that if this goes back out to review again, then I think we have no option but to retract the manuscript and go for another journal, otherwise we could be looking at a year and a half for publishing what is actually a very straight forward paper that hasn't really change much at all in that time.

Apologies if I sound frustrated, but this process has been exceptionally arduous.

If you have any news or queries or updates, please just email me on alan.jamieson@ncl.ac.uk

Regards

Alan